# NEDD9 Overexpression Causes Hyperproliferation of Luminal Cells and Cooperates with HER2 Oncogene in Tumor Initiation: A Novel Prognostic Marker in Breast Cancer

**DOI:** 10.3390/cancers15041119

**Published:** 2023-02-09

**Authors:** Marc L. Purazo, Ryan J. Ice, Rahul Shimpi, Mark Hoenerhoff, Elena N. Pugacheva

**Affiliations:** 1WVU Cancer Institute, School of Medicine, West Virginia University, Morgantown, WV 26505, USA; 2Unit for Laboratory Animal Medicine, University of Michigan, Ann Arbor, MI 48109, USA; 3Department of Biochemistry & Molecular Medicine, School of Medicine, West Virginia University, Morgantown, WV 26505, USA

**Keywords:** HER2, breast cancer, NEDD9 adaptor protein, hyperplasia, AURKA

## Abstract

**Simple Summary:**

Neural precursor cell expressed, developmentally downregulated protein 9 (NEDD9) is an adaptor protein that promotes integrin/RTK signaling. The overexpression of NEDD9 in HER2+ breast cancer correlates with reduced relapse-free survival. We generated a conditional transgenic mouse model of NEDD9 overexpression. When crossed with MMTV-Cre+ mice, NEDD9 upregulation led to mammary intraepithelial neoplasia (MIN) at an early age, and this phenotype was further exacerbated by Erbb2/(neu) oncogene. NEDD9 overexpression induces expansion of the mammary epithelial tree, with increased tertiary and terminal end buds (TEBs). The expression of NEDD9 increased the number of luminal epithelial cells, as shown by Keratin 8/Keratin 5 and Ki67 staining. Consistent with these studies, NEDD9 promoted the 2D proliferation and 3D mammary acini formation by normal human MCF10A cells. These findings support the role of NEDD9 in the early stages of HER2+ cancer, selectively impacting the proliferation of luminal epithelial cells, hence setting permissive conditions for tumorigenesis.

**Abstract:**

HER2 overexpression occurs in 10–20% of breast cancer patients. HER2+ tumors are characterized by an increase in Ki67, early relapse, and increased metastasis. Little is known about the factors influencing early stages of HER2- tumorigenesis and diagnostic markers. Previously, it was shown that the deletion of NEDD9 in mouse models of HER2 cancer interferes with tumor growth, but the role of NEDD9 upregulation is currently unexplored. We report that NEDD9 is overexpressed in a significant subset of HER2+ breast cancers and correlates with a limited response to anti-HER2 therapy. To investigate the mechanisms through which NEDD9 influences HER2-dependent tumorigenesis, we generated MMTV-Cre-NEDD9 transgenic mice. The analysis of mammary glands shows extensive ductal epithelium hyperplasia, increased branching, and terminal end bud expansion. The addition of oncogene Erbb2 (neu) leads to the earlier development of early hyperplastic benign lesions (~16 weeks), with a significantly shorter latency than the control mice. Similarly, NEDD9 upregulation in MCF10A-derived acini leads to hyperplasia-like DCIS. This phenotype is associated with activation of ERK1/2 and AURKA kinases, leading to an increased proliferation of luminal cells. These findings indicate that NEDD9 is setting permissive conditions for HER2-induced tumorigenesis, thus identifying this protein as a potential diagnostic marker for early detection.

## 1. Introduction

The HER2 (human epidermal growth factor receptor 2) amplification, or overexpression, occurs in 10–20% of breast carcinomas [1,2]. A high level of Ki67 expression, early relapse rate, and increased metastasis characterize HER2-positive (+) tumors [3]. The localized HER2+disease, confined to the breast tissue, has a 5-year survival rate of 97.3%, while regional disease with lymph node positivity is 82.8% and stage IV disease with metastases to distant organs is 38.8% [2]. This statistic highlights the importance of studies addressing the disease’s initiation to improve early detection and thus, survival.

Many advances have been made in treating HER2-positive breast cancer since the introduction of targeted monoclonal antibody therapies [4]. These include trastuzumab, pertuzumab-antibody-based therapies, and lapatinib, neratinib, and pyrotinib—the kinase inhibitors. However, many patients become resistant to treatment and relapse [5,6]. Identifying novel therapeutic targets is critical to combat resistance and improve outcomes.

HER2 (also known as v-erb-b2 avian erythroblastic leukemia viral oncogene homolog 2 (Erbb2) and neu) normally functions as a receptor tyrosine kinase (RTK) that can be activated by dimerization with other EGFR family members. Phosphorylation of the intracellular domain at (Tyr1144, Tyr1201, Tyr1226/1227, or Tyr1253) enables signaling through MAPK, AKT, and JNK, resulting in increased proliferation and survival [1,3]. Minimal increases in levels of HER2 allow for the altered growth of mammary glands [4], setting the stage for early breast lesions, leading to ductal carcinoma in situ (DCIS) [5,6]. Disease progression correlates with increased HER2 expression. Mouse models have been developed to analyze mammary gland transformation using rodent neu protein [7]. A MMTV-neu mouse model, expressing an unactivated form of rat Erbb2 gene (homolog of HER2), is similar to human cancers, characterized by long latency (29–48 weeks). This model allows for the analysis of early transformation events [8], starting from hyperplasia/dysplasia and moving toward adenocarcinoma [9]. Long latency indicates that additional events are required to enable transformation, since unactivated Erbb2 is primarily found in a monomeric/inactive form [10]. Little is known about early cooperating events in HER2-amplified tumors. The mutations in the p53 tumor suppressor, upregulation of heterodimerization partners, such as HER3 and EGFR, and activation of downstream targets, such as Src kinase, were often found in HER2+ breast cancer patients and promoted tumorigenesis [11].

NEDD9 (neural precursor cell expressed, developmentally downregulated 9) is an adaptor protein highly expressed in epithelial cells, and is critical for the activation of multiple kinases, including Src [12], FAK [13], AURKA [14], and integrins [8]. NEDD9, also known as Cas-L or HEF1, is a member of the CAS (Crk-associated substrate) family of adaptor/scaffold proteins that function in cell signaling [9]. Other CAS proteins, such as p130Cas, are also expressed in HER2+ breast cancer [15]. As shown by whole-body gene knock-out, endogenous NEDD9 is required for tumor growth in MMTV-neu transgenic mice [16]. NEDD9 is involved in cell division, migration, and invasion regulation, identifying this Cas-L family member as a potential cooperating oncogene in HER2-driven carcinogenesis [17,18,19,20]. However, specific processes affected by NEDD9 during the early stages preceding transformation are currently not well defined.

Here, we report that increased NEDD9 expression tightly correlates with the expression of HER2 protein in breast cancer patient biopsies and decreased anti-HER2 therapy response. Additionally, tissue microarray-based survival analysis shows that high expression of NEDD9 correlates with poorer overall survival. The conditional knock-in mouse model of NEDD9 in the mammary gland is characterized by mammary epithelial hyperplasia and promotes the early onset of Erbb2/Neu-driven tumorigenesis. Mechanistically, NEDD9 upregulation led to increased branching morphogenesis and expansion of luminal epithelial cells. In HER2+ human breast cancer cell lines, NEDD9 was upregulated compared to the control, and was correlated with proliferation. The upregulation of NEDD9 also correlates with a limited response to anti-HER2-targeted therapies, suggesting its involvement in sustainable downstream signaling. Our findings suggest that NEDD9 plays a crucial role in HER2+ breast cancer initiation, progression, and drug response; thus, targeting NEDD9 might provide new treatment strategies for BC patients and expose new vulnerabilities that could improve patient survival.

## 2. Materials and Methods

### 2.1. Generation of Conditional NEDD9 Knock-In (Floxed-STOP-NEDD9) Transgenic Strain and Locus Insertion

The targeting vector was utilized for homologous recombination at the Rosa26 locus of the murine genome. The cDNA of human NEDD9 (OriGene Technologies, Inc., Rockville, MD, USA) was cloned under the CAAG ubiquitously expressed promoter (CMV immediate early enhancer/chicken b-actin promoter fusion), separated by a stop cassette that was flanked by LoxP Cre recombinase recognition sites. GenOway, Inc., France, performed vector design, mouse embryo injection, and transplantation into a C57BL/6 background. This PCR is performed using a reverse primer (GX6044: 5′GCAGTGAGAAGAGTACCACCATGAGTCC3′) hybridizing in an exogenous sequence, added just upstream of the CAGG promoter, and a forward primer (GX6043: 5′AAGACGAAAAGGGCAAGCATCTTCC3′) hybridizing upstream of the targeting vector homology sequence.. Because of its localization, this primer set allows unequivocal and specific detection of the Rosa26 knock-in locus. Once homozygous floxed NEDD9 knock-in mice were generated, the transgenic line was cryopreserved and stored at the Jackson Laboratory, Bar Harbor, ME, USA. Due to the significant impact of genetic background on tumorigenesis, the NEDD9 transgene was transferred to the FVB/J background using speed congenic technology executed by Charles Rivers Laboratories (Wilmington, MA, USA) and WVU Transgenic Animal Facility. The successful transfer of NEDD9 transgene was confirmed by PCR analysis, as outlined.

### 2.2. Generation of FVB/J-MMTV-Cre-NEDD9+/+ Mice for Mammary Gland-Specific Upregulation of NEDD9

The FVB/J-Floxed-STOP-NEDD9^fx/fx^ mice were crossed with FVB/J-MMTV-Cre mice, expressing Cre recombinase under the mouse mammary tumor virus (MMTV) promoter, specific for mammary gland epithelium. This line was obtained via a gracious gift from Dr. J. Michael Ruppert (West Virginia University, Morgantown, WV, USA) [21]. The resultant offspring were genotyped using a set of designed primers to detect wild type, recombined, and Cre-induced (stop cassette excised) NEDD9 cDNA. Homozygous FVB/J-MMTV-Cre-NEDD9+/+ mice were used to analyze normal mammary gland development, and for other crosses with mammary tumor models. The upregulation of NEDD9 in the mammary epithelium was confirmed by Western blot analysis and immunohistochemistry.

### 2.3. PCR Screening Strategy for the Genotyping of the Homozygous NEDD9^fx/fx^ Knock-In Mice

The homozygous animals were identified using a PCR specific to the wild type Rosa26 allele. The primer pair, 027-ROSA-GX2942/028-ROSA-GX2943, has been designed and validated by GenOway for the specific detection of the wild type allele, producing 304 bp product. The forward primer 027-ROSA-GX2942 (5′CAATACCTTTCTGGGAGTTCTCTGC3′) hybridizes within the 5′ homology arm. The reverse primer 028-ROSA-GX2943 (5′CTGCATAAAACCCCAGATGACTACC3′) is located within the Rosa26 locus in a region, which is deleted in knock-in mice, and thus, no PCR product is produced.

### 2.4. PCR Screening Strategy for the Genotyping of the Induced NEDD9 Knock-In Line (With Removed Transcription Stop Cassette)

Mouse DNA was isolated from 0.3–0.5 cm tail fragments using the PrepEase Genomic DNA Isolation Kit (Affymetrix Inc., Cleveland, OH, USA), as per the manufacturer’s instructions. This PCR was performed using 61976cre-PUG1 (5′CTGCTCTATGACGGTCAGGATGTCTCC3′) and 61977cre-PUG1 (5′GGGCAACGTGCTGGTTATTGTGC3′) primers, enabling the detection of the Cre-mediated excised Rosa26 knock-in allele. The primer 61977cre-PUG1 is located within the NEDD9 transgene. The primer 61976cre-PUG1 hybridizes within the CAAG promoter. The Cre-excised recombined allele produced 296 bp product, while the non-excised recombined allele produced 3314 bp.

### 2.5. PCR Screening Strategy for the Genotyping of the Homozygous NEDD9/Cre/Erbb2 Mice

The FVB/J-MMTV-Cre-NEDD9+/+ mice were crossed with a FVB/J-MMTV-Erbb2/neu mouse model for mammary tumors in humans, purchased from The Jackson Laboratory (Cat#002376), Bar Harbor, ME, USA. This strain expresses unactivated, wild type cDNA of rat Erbb2 (neu). The resultant strain expressing all three transgenes, Cre, NEDD9, and Erbb2 (neu), was confirmed by PCR analysis using target-specific primers, as outlined above. The triple transgene female mice, along with controls, MMTV-Cre, MMTV-Cre-NEDD9+/+, and MMTV-neu, were used in studies outlined here. Mice were genotyped using the following primers: human NEDD9 cDNA: 5′TCCCAGAGTGTGCCGAGGAA3′, 5′GGGCCTTTTGCTGATGAGGG3′; ROSA: 5′CAATACCTTTCTGGGAGTTCTCTGC3′, 5′CTGCATAAAACCCCAGATGACTACC3′; Erbb2: 5′CGGAACCCACATCAGGCC3′, 5′TTTCCTGCAGCAGCCTACGC3′; Cre: 5′GGTTCTGATCTGAGCTCTGAGTG3′, 5′CATCACTCGTTGCATCGACCG3′. Using PCR RANGER mix (Meridian BioScience Inc., Memphis, TN, USA) resulted in the following band sizes, an indication of the presence of the NEDD9 (knock-in): 553 bp (base pairs), ROSA (wild type) 304 bp, Cre 900 bp, Erbb2 600 bp.

### 2.6. Generation of Mouse Embryonic Fibroblasts with NEDD9 Knock-In

Mouse embryonic fibroblasts (MEFs) were generated from embryos harvested at day 12–14 (E12–14), as previously described [22,23]. Tissue was minced using #10 scalpel blades and incubated for 15 min in 0.25% Trypsin-EDTA, at 37 °C. Full medium, Dulbecco’s modified Eagle’s (DMEM) medium, supplemented with 15% fetal bovine serum (FBS) and antibiotic-antimycotic, was added to neutralize the trypsin. Cells were pelleted by centrifugation, resuspended in a complete medium, plated, and passaged. All cell culture reagents were purchased from ThermoFisher Scientific, Waltham, MA, USA. Cells were cryopreserved for later use. As per manufacturer instructions, MEF cells were infected with adenovirus expressing GFP-tagged Cre-recombinase (green fluorescent protein) and Ad-Cre-GFP (Vector Biolabs, Malvern, PA, USA). Control and Ad-Cre-GFP positive cells, 72 h post-infection, were lysed and probed by Western blotting for NEDD9 and GAPDH protein levels, as previously described [24].

### 2.7. Quantification of Branching Density in Mammary Gland Whole Mounts

The fifth inguinal mammary gland was removed from the mouse, according to [25], and placed on a positively charged slide (StatLab, McKinney, TX, USA). It was subsequently fixed in Carnoy’s solution (Fisher Scientific, Pittsburgh, PA, USA) overnight and stained in a Carmine solution, prepared as previously described. Following staining, tissue was dehydrated with increasing ethanol, 70–100%, for 15 min each. To clear, tissue was placed in xylene for 12–72 h. Once the tissue became transparent, it was mounted using Permount Mounting Medium (Fisher Scientific, Pittsburgh, PA, USA). Images were captured using an Olympus VS120 Slide Scanner microscope (Olympus Lifescience, Center Valley, PA, USA) with 10×U Plan S Apo/0.75 NA objective. Quantification of mammary gland branching, and terminal end buds, was conducted, as previously described [25], using five mice per genotype and five randomly selected standard size fields of view per mouse.

### 2.8. Cell Culture, Plasmids, and Other Reagents

Cell lines BT-474, SK-BR-3, AU565, and MCF10A were purchased from and authenticated by American Type Culture Collection (Manassas, VA, USA), and were cultured based on the manufacturer’s recommendations. JIMT-1 was purchased and authenticated by Addexbio (San Diego, CA, USA). JIMT-1-Br3 was generously provided by Dr. Paul Lockman (West Virginia University School of Pharmacy, WV, USA) [26]. MCF10A-HER2-WT and MCF10A-Empty control were generously provided by Dr. Yehenew Agazie (Department of Biochemistry, West Virginia University School of Medicine, WV, USA) and were grown in medium, as previously described [27]. The cell lines were authenticated every 10–20 passages, and only low passage cells (2–6) were used for experiments. Cell medium supplements, including horse serum, EGF, penicillin and streptomycin, antibiotic/antimycotic, TrypLE, and trypsin, were purchased from Thermo Fisher Scientific, Waltham, MA, USA. FBS (fetal bovine serum) was purchased from VWR (Radnor, PA, USA). Insulin, cholera toxin, and hydrocortisone was purchased from Sigma-Aldrich (Sigma-Aldrich, St. Louis, MO, USA).

### 2.9. Lentivirus Constructs, Cell Infection, and Transfection Reagents

The cDNA for mouse NEDD9 (OriGene Technologies, Inc., Rockville, MD, USA) was subcloned (XhoI/EcoRI) into pLUTz (tet-inducible) lentivirus vectors [28], a gift from Dr. A. Ivanov (West Virginia University School of Medicine, WV, USA), to produce pLUTz-ms-NEDD9. The construct was transfected into Lenti-X™ 293T cells (Takara Bio, San Jose, CA, USA), based on the manufacturer’s recommendations, using a Turbofect transfection reagent (ThermoFisher Scientific, Waltham, MA, USA). The lentivirus particles were prepared for infection using established protocols [29], concentrated using ultracentrifugation, and 100 ul (~10^8^ MOI) was added to 50% confluent cells twice a day, for 3 days, with polybrene 5–10 ug/mL (Sigma-Aldrich, St. Louis, MO, USA). Infected cells were subjected to zeocin (InvivoGen US, San Diego, CA, USA) selection (200 ug/mL) for 2 weeks, with medium changed every three days.

### 2.10. Tumor Tissue Micro Array (TMA) and Patient Data

The Cancer Genome Atlas (TCGA, https://www.cancer.gov, accessed on 20 December 2022) breast cancer data and Kaplan-Meier Plotter (https://kmplot.com, accessed on 20 December 2022) web-based tools were used to evaluate genomic DNA and RNA-seq expression of NEDD9 in HER2+ breast cancer, as previously described [30,31]. High-density breast cancer tissue microarrays BR2082 and BR1008 were used in this study. Diagnostics, stage, and HER2 positivity scoring data are provided in Appendix A. The biopsies were collected with full donor consent by US Biomax Inc./TissueArray Derwood, MD, USA. The samples in the TMA were selected to represent two groups: 1. nonmalignant (NM) biopsies that include normal tissue from healthy donors, normal adjacent to the tumor tissue, hyperplasia, and benign lesions; 2. malignant HER2+ biopsies, including carcinomas, invasive ductal carcinomas, and lymph node metastases. In total, 90 samples were analyzed, including 6 normal breast tissue, 10 normal adjacent to tumor tissue, 16 hyperplasia lesions, 8 benign lesions, 36 malignant tumor samples, and 14 lymph node metastases.

### 2.11. Immunohistochemistry and Scoring Procedures

Immunohistochemistry (IHC) was done according to the manufacturer’s recommendations (US Biomax Inc. Derwood, MD, USA), using previously validated anti-NEDD9 antibodies (clone 2G9) [32] and HER2. Manual scoring of staining intensity [negative (0), weak (1+), moderate (2+), or strong (3+)], as well as location and cell types, was completed by an independent pathologist from US Biomax, Inc. Each core was scanned by the Aperio Scanning System (Leica Biosystems, Deer Park, IL, USA), as previously described [24]. The total number of positive cells and the intensity of NEDD9 staining were computed by Aperio Image- Scope10.1 software, based on the digital images taken (×20) from each core, and normalized to the background control and normal adjacent tissue.

### 2.12. Fluorescent Immunohistochemistry and Hematoxylin & Eosin (H&E) Staining

Briefly, deparaffinization and rehydration of 5 μm sections were performed as follows: 1. three incubations for 3 min in xylene; 2. three incubations for 2 min in 100% ethanol; 3. 2 min each in 95, 80, and 70% ethanol; 4. 5 min in 1× TBS (Tris-Buffered Saline). All chemicals, BSA, and buffers were purchased from Fisher Scientific, Pittsburgh, PA, USA. Antigen retrieval was performed using a citrate buffer, pH 8.0 at 98C, for 20 min, as previously reported [32]. Sections were subsequently blocked using 5% Bovine Serum Albumin (BSA), 1× TBS solution. Sections were stained with anti-Ki67-AlexaFluro-555 conjugated antibody (BD Biosciences, Franklin Lakes, NJ, USA), anti-cytokeratin 8 [33] (1:100 dilution, TROMA-I was deposited to the DSHB (Developmental Studies Hybridoma Bank) by Drs. Brulet, P. & Kemler, R.), and anti-cytokeratin5 (Poly19055, BioLegend San Diego, CA, USA), dilution 1:500, and incubated overnight, as previously reported [34]. Secondary antibodies that were diluted, 1:10,000, included AlexaFluor-488, 555, and 647 (ThermoFisher Scientific, Waltham, MA, USA). The sections were mounted with ProLong Gold DAPI-containing media (ThermoFisher Scientific, Waltham, MA, USA). Images were captured using an Olympus VS120 Slide Scanner microscope with 10× U Plan S Apo/0.75 NA objective (Olympus Lifescience, Center Valley, PA, USA).

For H&E, staining slides post deparaffinization and rehydration were incubated with hematoxylin (Fisher Scientific, Pittsburgh, PA, USA) for 45 s, rinsed with water for 30 s, followed by Eosin for 1 min, rinsed and dehydrated in 95% ethanol, followed by three incubations for 1 min in 100% ethanol and three incubations for 1 min in xylene. Slides were mounted using Richard-Allan Scientific Mounting Medium (ThermoFisher Scientific, Waltham, MA, USA). Samples were evaluated by an experienced pathologist blinded to the identity of the slides.

### 2.13. Histopathology Evaluation and Grading

Histopathology assessments were completed by a board-certified veterinary anatomic pathologist (MJH) blinded to the identity of the samples. Hematoxylin and eosin-stained slides were used to assess the mouse mammary glands (4th inguinal gland was used) for MIN, DCIS, and mammary tumors (adenoma, carcinoma). The slides were digitally scanned at 40× magnification using an Aperio slide scanner, and the total number of lesions were counted per mammary gland. A total of 32 slides from 16-week-old females with four genotypes (Cre, Cre-NEDD9, Cre-neu, Cre-neu-NEDD9) were evaluated. The average area per slide was 100–120 mm^2^. Low-grade MIN lesions were characterized by glands lined by 1–2 layers of atypical epithelial cells with enlarged nuclei, clumped chromatin, variable nuclear size and shape, and an increased mitotic rate. High-grade MIN lesions were typified by loss of lumens and a thickening of the epithelium, by multiple layers of disorganized epithelial cells, with increased pleomorphism. Ductal proliferative lesions (DCIS) were observed as a marked expansion and filling of mammary ducts with a monomorphic population of epithelial cells, causing marked enlargement of the gland, without invasion of the adjacent basement membrane or formation of a palpable mass. DCIS lesions can be considered high-grade MIN lesions [35,36].

### 2.14. Western Blotting

Western blotting was performed using standard procedures [24]. The primary antibodies used were anti-NEDD9 (2G9), -phospho-ERK1/2(D13.14.4E), -ERK1/2(137F5), -phospho-Aurora Kinase A(D13A11), -HER2 (29D8), -phospho-Src (100F9), FAK (D2R2E) (Cell Signaling Technology, Danvers, MA USA), anti-phospho-FAK (Tyr397) (ThermoFisher Scientific, Waltham, MA, USA), anti-Src (GD11) (Sigma-Aldrich, St. Louis, MO, USA), anti-Aurora Kinase A (BD Biosciences, Franklin Lakes, NJ, USA). The dilution was based on the manufacturer’s recommendation. Secondary anti-mouse and anti-rabbit horseradish peroxidase (HRP)-conjugated antibodies (Jackson ImmunoResearch Laboratories, Inc., West Grove, PA, USA) were diluted, 1:10,000, followed by chemiluminescence-based detection with Luminata Forte Western HRP substrate (Sigma-Aldrich, St. Louis, MO, USA), and were quantified using ImageJ software (ImageJ—National Institute of Health, https://imagej.nih.gov, accessed on 12 December 2022).

### 2.15. MTT-Based Proliferation and Viability Assay

Cell Proliferation and IC50 values were assessed via MTT (ThermoFisher Scientific, Waltham, MA, USA) [3-(4, 5-dimethylthiazol-2-yl)-2, 5-diphenyl tetrazolium bromide] assay. Briefly, 1 × 10^5^ cells/well were plated in a 96-well plate. At 24, 48, 72, and 96 h time points, 10 uL of MTT (5 mg/mL) was added to each well, incubated at 37 °C for 4 h, then dimethyl sulfoxide (DMSO)was added, and absorbance was measured using the Cytation5 imaging system (Agilent Technologies, Santa Clara, CA, USA) at 540 nm. The results were reported as a fold of change over control at each time point.

### 2.16. Drug Treatment and Viability Assay

Cells were plated at 2.5 × 10^3^, 5.0 × 10^3^, and 10 × 10^3^ density, per 96-well plate, in triplicates. For each cell line, two different shRNA targeting NEDD9 (ThermoFisher Scientific, Waltham, MA, USA) and scramble shRNA non-targeting controls were used, as previously described [24]. Cells were treated with various concentrations of lapatinib (ThermoFisher Scientific, Waltham, MA, USA) for 48 h. CyQUANT Cell Proliferation Assay (ThermoFisher Scientific, Waltham, MA, USA) was utilized, per the manufacturer’s protocol, to quantify the number of viable cells per well using Biotek Cytation Plate Reader (Agilent Technologies, Santa Clara, CA, USA) excitation/emission 480 nm/520 nm. Percent viability was calculated by setting the control cell line to 100%. One-way ANOVA was used to calculate statistical significance in two independent experiments with three technical replicas.

### 2.17. Acini Formation, Imaging, and Quantification Procedures

The Acini/Morphogenesis assay was performed as previously described [37]. Briefly, 10^5^ cells/mL cells were resuspended in DMEM/F12 medium, supplemented with 2% horse serum, 10 μg/mL insulin, 1 ng/mL cholera toxin, 100 μg/mL hydrocortisone, 1× antibiotic-antimycotic. Eight-well cell culture chamber slides (ThermoFisher Scientific, Waltham, MA, USA, USA) were coated with 45 ul of Matrigel (ThermoFisher Scientific, Waltham, MA, USA, USA) per well. The cells were mixed 1:1 with DMEM/F12 medium, containing 4% Matrigel, and were added to a Matrigel-coated slide. The medium was replaced every 3–4 days. The acini formation was documented daily via microscopy for a total of 10–14 days. The images from five randomly selected fields of view of standard size were collected with 10× U Plan S Apo/0.75 NA objective using Echo Rebel microscope (Discover Echo, San Diego, CA, USA). After 14 days, acini were fixed, and immunofluorescent analysis was conducted, as previously described [24]. Briefly, acini were fixed using 3% paraformaldehyde for 20 min, followed by washing with 1× PBS (Phosphate Buffered Saline). Permeabilization, using 1× PBS containing 0.5% Triton X-100, was conducted for 10 min at 4 °C, and then rinsed three times using 1× PBS/Glycine for 10 min each. Acini were mounted with ProLong Gold DAPI-containing media (ThermoFisher Scientific, Waltham, MA, USA, USA). A total of 10 fields with acini per condition were imaged using Ti2Nikon Spinning Disk Confocal microscope (Nikon Instruments Inc. Melville, NY, USA), equipped with CSU-W1 Yokogawa spinning head (Yokogawa America, Sugar Land, TX, USA); z-stack of images taken at 10 mm per step, up to 200 mm total, were processed using NIS software, and 3D projection images were quantified. The total number of cells per acini was quantified using DAPI. Bright-field images were used to measure the size/diameter of the acini. The 50 total acini per cell line, in 10 randomly selected fields of view of standard size, were quantified in three independent experiments.

### 2.18. Kaplan–Meier Analysis and Drug Response Evaluation Using ROC Plotter

Kaplan–Meier (KM) analysis, to evaluate the prognostic value of NEDD9 in breast cancer, was completed using a publicly available microarray database from breast cancer patients, as outlined at https://kmplot.com/analysis/index.php?p=service&cancer=breast, accessed on 20 December 2022. The relapse-free survival (RFS) was calculated in a cohort of HER2-positive (n = 1273, HER2+ status–array). All patients, treated and untreated, were included. The ROC Plotter is the first online transcriptome-based validation tool for predictive biomarkers of therapy response (https://www.rocplot.org/, accessed on 20 December 2022) [38]. Most patients in the pCR (pathological complete response) and RFS cohorts have received chemotherapy. The neoadjuvant or adjuvant settings for chemotherapy were not defined. The small patient cohort used in our analysis was treated with lapatinib (n = 65) and trastuzumab (n = 186). Patients included in RFS ROC plots received anti-HER2 therapy in adjuvant settings. The following analysis parameters were used: Gene-NEDD9 (JetSet best probe set (202149_at)) and Gene-ERBB2 (JetSet best probe set (216836_s_at)). The HER2+ patients treated with any anti-HER2 therapy were split by the auto-select best cutoff, unless indicated otherwise in the text. The data were censored at the threshold; the redundant samples were removed. The biased arrays were removed; the *p*-value was calculated using the log-rank test and plotted in R. The specific parameters used are outlined in Appendix A.

### 2.19. Statistical Analysis

Statistical comparisons were made using a two-tailed Student’s *t*-test, or one-way or two-way analysis of variance (ANOVA), when more than two samples were compared. *p* ≤ 0.05 was considered to be significant (*). Experimental values were reported as the means with ± S.E.M (standard error of the mean), *p* values were reported as adjusted, and statistical significance calculations were made using Prism 9.0.0 software (GraphPad Software, Inc., San Diego, CA, USA). All experimental data sets reported here were collected from multiple independent experiments with multiple technical and biological replicas.

## 3. Results

### 3.1. NEDD9 Expression Correlates with HER2+ Disease Progression and Treatment Outcomes

NEDD9 expression was previously documented to correlate with poor prognosis in triple-negative breast cancers [24,30]. However, the impact of NEDD9 expression on HER2+ breast cancers (BCs) is currently unknown. The breast cancer tissue microarrays (TMAs), with breast patient biopsies, were analyzed for NEDD9 expression using a validated antibody to address this gap. The results of quantitative immunohistochemistry staining were then correlated with the pathological stage of the disease and the HER2+ score (0+, 1+, 2+, 3+) assigned by a pathologist (Figure 1A–D). When compared to a nonmalignant (NM) tissue, which includes normal adjacent tissue (NAT), hyperplasia and benign disease expression of NEDD9 increased from nonmalignant to invasive ductal carcinoma (IDC) and metastatic lesions (lymph node-Met/LN; Figure 1A-B). Moreover, analysis of NEDD9 expression, in the nonmalignant group, shows a significant increase in NEDD9 expression in benign and hyperplastic lesions when compared to NAT and normal tissue (Figure 1C; Appendix A), indicative of a positive correlation between disease progression and NEDD9 expression at earlier stages. The difference in NEDD9 expression, between patient cohorts with assigned HER2 pathological score 1+, 2+, or 3+, was non-significant (Figure 1D). The clinical relevance of these findings is supported by the in-silico analysis of publicly available microarray data from TCGA (Kaplan-Meier Plotter) [32,39]. The results show a strong correlation between NEDD9 mRNA and relapse-free survival (RFS) in HER2+ breast cancers (BCs). In HER2+ tumors, a higher expression of NEDD9 correlates with lower RFS (Figure 1E; HR = 1.29, *p* = 0.02). The RFS analysis of ERBB2 shows similar findings (Figure 1E). Higher expression of ERBB2 was associated with worsened RFS (HR = 1.31, *p* = 0.04). Contrary to the findings in HER2+ and TN BCs (HR-1.39, *p* = 0.0048), the RFS of estrogen receptor positive (ER+) BCs shows a positive correlation between increased NEDD9 expression and relapse free survival (HR = 0.73, *p* = 0.00014, Appendix A), indicative of different NEDD9 signaling. Overall, this data suggests that NEDD9 is elevated at the protein and RNA levels in HER2+ breast cancers, and that increased levels of NEDD9 correlate with lower RFS in the HER2+ subset of BC patients, but not in ER+ BCs. 

In agreement with low RFS, the NEDD9-high patients also demonstrated lower response rates to anti-HER2 therapy in HER2+ breast cancer patients (Figure 1F,G https://www.rocplot.org, accessed on 20 December 2022), as documented by lower pCR (pathological Complete Response) and RFS 5-year survival post treatment. The area under the curve (any anti-HER2 therapy, AUC (area under the curve) = 0.612, *p* = 0.0067; and AUC = 0.719, *p* = 0.004) criteria shows significant influence of NEDD9 expression on pCR and relapse-free 5-year survival. The graphs show better treatment outcomes when using anti-HER2 therapies in NEDD9-low patients. This trend remains true in HER2 breast cancers, classified based on molecular subtype, as defined by St. Gallen criteria [40]. Moreover, based on ROC plotter analysis, NEDD9 outperforms HER2/ERBB2 (probe#216836) in predicting anti-HER2-therapy response and survival post-therapy in HER2+BCs (any anti-HER2 therapy; pCR AUC = 0.539, *p* = 0.2; ns non—significant; RFS AUC = 0.658, *p* = 0.031). Trastuzumab was the only treatment with all breast cancer patients, included independently of HER2 status, that provided better correlation with HER2 expression (pCR AUC = 0.629, *p* = 0.00084). There was no correlation between NEDD9 expression and chemotherapy treatment (Appendix A). However, we found that NEDD9 expression in HER2+ BCs (but not HER2-negative BCs) is highly predictive of treatment outcomes in patients treated with FEC chemotherapy (Fluorouracil, Epirubacine, Cyclophosphamide; AUC = 0.75, *p* = 6.2 × 10^−11^; Appendix A). The NEDD9 expression does not influence RFS (AUC = 0.51, *p* = 0.35; Appendix A) or pCR response to endocrine therapy in ER+ breast cancers (AUC = 0.505, *p* = 0.47; Appendix A). Thus, understanding the role NEDD9 plays in HER2-driven BCs is critical for identifying patients with a higher risk of relapse and drug resistance.

### 3.2. Generation of Conditional NEDD9 Knock-In (KI) Transgenic Mouse Model

Previously, it was shown that deletion of NEDD9 via gene knock-out (KO), in mice overexpressing wild type (unactivated) Erbb2 (neu), leads to a drastic decrease in HER2-driven tumorigenesis, with only 10–15% of mice being able to form tumors in their lifetime. Note that this tumor incidence (~13.9%) corresponds to naturally occurring spontaneous mammary gland tumors documented in this strain [10], suggesting that NEDD9 is critical for the oncogenic activity of HER2, and without it, no tumors above naturally set background can arise. To determine the role of NEDD9 in HER2-induced tumorigenesis, we generated a NEDD9 overexpression transgenic mouse model to mimic pathological conditions observed in human HER2+ BCs.

The targeting vector containing a cDNA sequence for human NEDD9, under the CAG promoter (hybrid promoter consisting of the cytomegalovirus (CMV) enhancer fused to the chicken beta-actin promoter), was utilized for homologous recombination at the Rosa26 locus of the murine genome. The promoter was followed by a stop cassette flanked by LoxP Cre recombinase recognition sites. This design allows inducible, tissue/cell type-specific expression of an extra copy of NEDD9 (Figure 2A). The vector design and C57BL/6 mouse embryo injection/transplantation were carried out in collaboration with GenOway (Lyon, France). The PCR screening for homologous recombination at the 5′ end of the targeting vector was performed using a set of designated primers, as outlined in Materials and Methods. Next, the conditional (floxed, fx) NEDD9 transgene was transferred into the FVB background, using speed congenic services provided by Charles Rivers Laboratories (USA). The resultant homozygous strain FVB/J-LoxP-STOP-LoxP-NEDD9, called NEDD9fx/fx, was confirmed by PCR (Figure 2B) and Southern blotting. To test the effectiveness of the transgenic construct, mouse embryonic fibroblasts (MEFs) were isolated from NEDD9^fx/fx^ E14 embryos. The MEFs were infected with pre-packaged adenovirus, expressing Cre recombinase, fused with GFP (green fluorescent protein, Ad-Cre-GFP). The Western blot analysis of control (GFP only) and Cre-GFP infected cells for NEDD9 shows upregulation of its expression, thus confirming the functionality of inserted whole-body transgene upon Cre expression (Figure 2C).

### 3.3. Production and Analysis of Mammary Gland-Specific Expression of NEDD9

To assess the specific role of NEDD9 overexpression in mammary tumorigenesis and mammary gland development, we crossed the NEDD9fx/fx mice with a strain expressing Cre recombinase under the control of the mouse mammary tumor virus promoter, MMTV-Cre (Figure 2D) [21]. The offspring were genotyped (Figure 2E) using a set of primers described in Material and Methods. The resulting mice overexpressed NEDD9 protein in the mammary gland epithelial cells. Homozygous MMTV-Cre-NEDD9+/+ mice and appropriate controls (MMTV-Cre and NEDD9fx/fx) were used to analyze mammary gland development and were crossed with mammary tumor models. The upregulation of NEDD9 in the mammary epithelium was confirmed by Western blot and immunocytochemistry analysis (Figure 2F). Mice overexpressing NEDD9 in mammary epithelium developed normally and produced/nursed healthy offspring. In conclusion, we developed a mouse model of conditional NEDD9 expression that can be successfully used to upregulate NEDD9, in a tissue-specific manner, to enable further evaluation of NEDD9 function in vivo, under normal and pathological conditions.

### 3.4. NEDD9 Overexpression Alters Mammary Gland Architecture by Increasing Mammary Gland Budding and Branching Morphogenesis and Cooperates with HER2

Previously, it was shown that whole-body NEDD9 knock-out did not affect mammary gland development [16], while the impact of overexpression is currently unknown. To address this gap in our knowledge, we compared mammary gland architecture between MMTV-Cre and MMTV-Cre-NEDD9+/+ mature nulliparous mice. The whole mammary gland mounts were prepared from 16-week-old female mice (Figure 3A,B). We found that, upon NEDD9 overexpression, the buds-to-branch ratio increased. There were no significant differences between the groups in the secondary and tertiary branches (Figure 3C), while more terminal end buds (TEBs) were observed in NEDD9 overexpressing glands compared to MMTV-Cre (Figure 3D). The changes in the branch-to-bud ratio are indicative of an increased invasion and/or proliferation during mammary gland development. Thus, overexpression of NEDD9, in normal mammary epithelium, might lead to early hyperplasia.

### 3.5. Cooperation between NEDD9 and HER2 Promotes Mammary Gland Branching Morphogenesis

The MMTV-Cre-NEDD9+/+ and MMTV-Cre mice were further crossed with the MMTV-Erbb2/neu (JAX Strain #002376) model to assess the impact of NEDD9 upregulation on oncogene-driven carcinogenesis. The MMTV-neu model develops physiologically relevant, spontaneous mammary tumors [10]. Similarly to many human HER2+ breast cancers, it overexpresses the wild type form of Erbb2, which develops slowly, starting from foci of hyperplastic, dysplastic mammary epithelium and progressing to aggressive metastatic carcinoma [41]. The first lesions appear at 4–6 months, with a median incidence of 205 days. Both virgin and non-virgin female mice develop tumors. The MMTV-Cre-Erbb2 and MMTV-Cre-Erbb2-NEDD9^+/+^ mice were successfully produced, and nulliparous females were examined for mammary gland development and early precancerous\benign lesions at ~16 weeks of age. This time point was selected based on early preneoplastic lesions detected in Cre-NEDD9^+/+^ female mice (Figure 3A–D) and previously published reports on MMTV-Erbb2/neu mice [39]. We found that mice expressing MMTV-Cre-Erbb2-NEDD9^+/+^ had more tertiary branches and TEBs when compared to MMTV-Cre-Erbb2/neu mice (Figure 3E–H).

### 3.6. NEDD9 Overexpression Is Associated with Mammary Intra-Epithelia Neoplasia

In agreement with TEBs expansion, the analysis of mammary glands showed a significant increase in the number of mice with mammary intra-epithelial neoplasia (MIN) lesions associated with NEDD9 overexpression (Figure 4A,B, Table 1). The MIN was observed at ~16 weeks of age and characterized by glands lined by 1–2 layers of atypical epithelial cells, with variable nuclear size and shape. The loss of a noticeable lumen and thickening of the epithelium by multiple layers of disorganized cells, with increased pleomorphism and mitotic rates, are consistent with high-grade MIN lesions (Figure 4A). Since mammary intraepithelial neoplasia, including DCIS, is associated with an increased risk of breast cancer [32], these results suggest that NEDD9 upregulation might be pre-disposed to neoplastic transformation. Similarly, the MIN lesions were found at higher frequency in triple transgene mice with NEDD9 than Erbb2 alone (Figure 4B, Table 1). These findings suggest that overexpression of NEDD9 increases the frequency of preneoplastic events in the Erbb2 model. Thus, cooperation between NEDD9 and HER2 promotes the early initiation of tumorigenesis.

### 3.7. NEDD9 Overexpression Causes Hyperproliferation of Luminal Cells

The terminal end buds are responsible for the production of mature luminal and myoepithelial cells, leading to ductal tree formation [40,42]. To determine if TEB expansion and the development of the MIN lesions is due to increased proliferation, immunohistochemistry with Ki67 antibody was performed (Figure 5A,B). The upregulation of NEDD9 and Erbb2 proteins (Figure 5C,D) in the mammary gland led to an increased number of Ki67-positive cells in ducts normalized to the total number of cells (Figure 5A,B). The Ki67 positivity correlates with increased expression of NEDD9 and HER2 (Figure 5C,D) in these cells, suggesting cooperation. Next, the immunofluorescent staining with antibodies against known markers of luminal epithelial (cytokeratin 8) and basal myoepithelial (cytokeratin 5) cells was conducted (Figure 5E,F). The NEDD9 overexpressing mice had a significantly increased number of luminal epithelial cells (Figure 5E,F). No significant differences in myoepithelial cells (Figure 5E,F) were observed. These data agree with a previously published report documenting a reduction in luminal progenitors and mature cells in NEDD9 knock-out mice [16]. Overall, these results suggest that NEDD9 plays a crucial role in luminal cell proliferation and expansion of the ductal compartment, which is a primary source of human cancer.

### 3.8. NEDD9 Is Overexpressed in Human HER2+ Breast Cancer Cell Lines

Similarly to TMA findings (Figure 1), NEDD9 protein was increased in a panel of HER2+ breast cancer cell lines (BT-474, SK-BR-3, JIMT1, and JIMT1-Br3) when compared to non-transformed MCF10A cells (Figure 6A,B). Next, we tested the impact of NEDD9 expression on sensitivity to lapatinib, which is a standard of care HER2-targeting drug. The HER2+ breast cancer cell lines that have been previously reported as sensitive (SKBR-3, AU565) or resistant to lapatinib (JIMT-1) [43] were used. In agreement with ROC data (Figure 1E), depletion of NEDD9 by shRNAs (Figure 6C) in resistant cells—JIMT1—improved the response to lapatinib. Similarly, a decrease in NEDD9 led to increased sensitivity to lapatinib in AU549 and BT474 cells (Figure 6D), suggesting that the combination of anti-HER2 therapies with NEDD9 depletion might lead to improved outcomes.

### 3.9. NEDD9 Overexpression in Normal Mammary Epithelial Cells Causes Increased Proliferation

To evaluate our findings on NEDD9-driven MIN/DCIS increase in the human model, we overexpressed NEDD9 in human mammary epithelial cells—MCF10A—alone, or in combination with HER2 or empty vector controls (Figure 7A). We found that MCF10A cells overexpressing NEDD9 significantly increased proliferation in a 2D assay after 72 h, suggesting that NEDD9-driven release from contact inhibition normally restrains proliferation (Figure 7B). The cells grown in a 3D Matrigel matrix form duct-like structures (acini)—polarized spheroids that consist of a single layer and have a hollow center. It was previously shown that, upon HER2 upregulation/activation, the acini enlarge via excessive proliferation, loss of apical polarity, and lack apoptosis [39]. A similar phenomenon was observed in MCF10A cells that overexpressed exogenous NEDD9 alone (Figure 7C,D), confirming the critical role of NEDD9 in the regulation of proliferation in non-transformed cells. There was no significant additive effect in acini growth in combination of NEDD9 with HER2, since additional changes are required to enable HER2 receptor engagement [8]. The experimental window of 10–14 days used for acini formation is insufficient to allow for such changes to happen in comparison to mice studies. It was shown that NEDD9 interacts with and activates the mitotic kinase AURKA [32]. Hence, we analyzed the activation of AURKA in our experimental model. The level of active, phosphorylated AURKA was significantly increased upon NEDD9 overexpression. Interestingly, overexpression of HER2 stabilized total AURKA, but did not increase its activity. The phosphorylation of ERK1/2 was also significantly increased (Figure 7E) upon overexpression of NEDD9, indicative of MAPK signaling activation. The SRC and FAK—classical targets of HER2 downstream signal transduction—were not significantly affected in MCF10A-NEDD9 cells compared to the control (Figure 7F), suggesting non-canonical AURKA-driven activation of proliferation in human cells with upregulated NEDD9.

## 4. Discussion

The HER2-positive breast cancer subtype shows a high expression of the HER2 kinase and is associated with proliferation-related genes, characterized by high K67/mitotic index and aggressive growth [44]. HER2 signaling function is enabled via ligand-bound heterodimerization, with HER3 or EGFR [45], that leads to phosphorylation and docking of multiple signaling adaptor molecules, thus activating several downstream pathways, including proliferation (ERK1/2), invasion (FAK/Src), and survival (AKT) [46]. A few adaptor molecules were documented to be key signaling hubs in HER2 oncogenic activity, including Grb2 [47], p130Cas [15], and NEDD9 [16]. The knock-out of murine Nedd9 led to a reduction in luminal progenitors, HER2 signaling, and tumor growth [16]. While deletion of NEDD9 is not observed in human breast cancers, the upregulation of NEDD9 has been linked to tumor progression and metastasis in various cancers, including breast, liver, colon, pancreatic, ovarian, lung, and brain [18,19,20,48,49]. However, the impact of the upregulation of NEDD9 on HER2+ breast cancers has not been explored. Here, we report that NEDD9 protein (Tissue MicroArray (TMA), Figure 1A–C and Figure 6A–C) and mRNA (TCGA, Figure 1D) expression Is elevated in HER2+ human breast cancer patient biopsies and established cell lines. TMA analysis was conducted to evaluate NEDD9 expression in HER2+ breast cancers (BCs), with variable HER2 expression (score), and to compare it with nonmalignant stages and metastatic lesions. It is well established that NEDD9 expression correlates with the onset of EMT and increased migration/invasion and metastasis [24,48]. Our group has previously reported the expression of NEDD9 in triple-negative breast cancers (TNBCs). Other groups have reported on the NEDD9 in ER+ breast cancers. Little is known about the role NEDD9 plays in HER2+ BCs and the early stages of tumorigenesis. The analysis of NEDD9 expression across nonmalignant groups shows a significant increase in NEDD9 levels, as early as hyperplasia and benign lesions, which supports our conclusions. The NEDD9 upregulation correlates with disease progression, HER2 expression, low anti-HER2 therapy response, and relapse free survival (RFS). Further clinical research is needed to utilize benign biopsies to evaluate NEDD9 as a potential biomarker of disease progression in clinical settings, which could enable early diagnostics and the identification of patients at high risk for cancer at early stages.

To determine the impact of NEDD9 on HER2 carcinogenesis and disease progression, we generated a mouse model overexpressing NEDD9, specifically within the mouse mammary gland. We analyzed the effect of NEDD9 upregulation in MMTV-neu (rodent homolog of HER2, unactivated). The MMTV-neu mice form tumors spontaneously at ~29–48 weeks [39]. The long latency is thought to be due to the need for additional mutations to enable transformation, including p53, which is often observed in HER2+ breast cancers [50]. The upregulation of NEDD9 resulted in higher rates and earlier formation of preneoplastic, benign lesions, such as mammary intraepithelial neoplasia (MIN) and DCIS. These findings suggest that the upregulation of NEDD9 might play a key role in the transition from normal to preneoplastic and neoplastic lesions [42].

To better understand the effects of NEDD9 overexpression within the MMTV-neu mouse model, we evaluated the mouse mammary gland for changes, specifically at the morphological and cellular levels. NEDD9 overexpression was able to increase the number of tertiary branches, as well as terminal end buds, which suggested NEDD9 plays a role in proliferation. The expansion of the ductal tree and proliferation at the leading edge of the terminal end buds point to changes in mammary gland development [42]. We found that the proliferation of luminal (cytokeratin 8+) cells was increased, while the number of myoepithelial cells (Keratin 5+) was not significantly affected. There was a significant increase in double positive K5+/K8+ cells. The normal breast tissue contains subpopulations of mammary stem cells (MaSC) and progenitors. The origin of different BC tumor subtypes often correlates with one of the subgroups. The basal-like and HER2+ BCs resemble the luminal progenitor cells. These cells are often dual-positive KRT5+/KRT8+ and are found in the luminal layer of the duct [51]. In agreement with our findings in NEDD9 transgenic mice, the HER2-enriched cell lines BT474 and SKBR3 showed expression of both keratin 5 and 8 [51]. The increase in double-positive luminal cells supports the conclusion that NEDD9 overexpression increases the pool of cells, considered a cell of origin, for transformation in the HER2+ subtype. The knock-out of murine NEDD9 in the MMTV-neu model was shown to reduce the number of luminal progenitors [16], but did not affect mammary gland architecture. In agreement with earlier findings, the overexpression of NEDD9 led to the expansion of luminal cells.

The increase in stem-like population, in combination with increased proliferation, might explain the increase in the incidence of benign lesions upon NEDD9 overexpression. NEDD9 promotes stemness across various cancers [16,52,53]. Cancer stem/initiating cells were found in the patient’s biopsies and significantly influenced the risk of relapse [54,55]. In this context, NEDD9-dependent increases in initiation, and RFS might be connected. Similarly, the role of NEDD9 in EMT, previously documented by multiple reports, links it to resistance to therapies. Though no NEDD9-targeting drugs are currently available, the inhibitors for some of its downstream effectors, like AURKA and FAK, are available and shown to be effective [56,57].

To evaluate our findings in the context of human cancer, we analyzed the expression of NEDD9 in multiple HER2+ human breast cell lines. The protein expression of HER2 shows some variability among cell lines, with the lowest expression in JIMT1, HER2-therapy drug-resistant cells. Similarly, four of the five HER2+ cell lines had increased expression of NEDD9 protein when compared to non-transformed MCF10A cells. Although AU565 and SK-BR-3 are isogenic cell lines isolated from the same patient and have a similar mutation landscape, like p53 mutant (p53 R175H), they vary in HER2/NEDD9 levels and differ in downstream signaling. Unlike SK-BR-3, AU565 is poorly responsive to EGF and has higher HER3 expression that might enable direct MAPK/AKT activation without NEDD9 [58]. To further explore the effect of NEDD9 upregulation on non-transformed breast epithelial cells, we overexpressed both proteins in MCF10A cells. The parental cell line has low levels of NEDD9 and HER2. The 2D proliferation assay shows an increase in proliferation upon overexpression of NEDD9 alone, or in combination with HER2. Similarly, the MCF10A-NEDD9 and MCF10A-NEDD9-HER2 cells, when grown in a 3D Matrigel matrix, formed significantly larger acini, as documented by an increased diameter and the number of cells. NEDD9 cells have an enhanced ability to continue proliferation, even if confluence is achieved. Mechanisms, such as contact inhibition as restriction of proliferation, do not occur and NEDD9 is able to overcome such limitations. Taken together, our results suggest that NEDD9 might serve as a predictive biomarker of preneoplastic, an early stage in HER2-driven carcinogenesis.

Recent studies confirm that the HER2 molecular subtype yields the best clinical and therapeutic response by anti-HER2 therapies. Resistance to anti-HER2 therapies has been studied at great length. Some current factors of resistance are: 1. changes in the binding sites or to tyrosine kinase receptor domain; 2. overexpression of HER2 receptor; 3. dimerization with other receptors for activity; 4. alternate activation of downstream signaling pathways [59]. Increased NEDD9 expression correlates with non-responders. Area under the curve (AUC) findings, across different BC subtypes (Figure 1, Appendix A), indicates that NEDD9 is a strong prognostic biomarker of anti-HER2 and FEC therapy response in HER2+ BC patients. The AUC data show no correlation between NEDD9 expression and response to tamoxifen in ER+ BCs [58]. Furthermore, elevated expression of NEDD9 also serves as a key indicator of relapse free survival (RFS); thus, it can be used as a marker of possible disease reoccurrence.

## 5. Conclusions

A few adaptor molecules were reported to be critical downstream effectors of the HER2 oncogene. In this study, we outline the key role of the Cas-family scaffolding protein—NEDD9—in the initiation and progression of HER2-driven tumors. NEDD9 is elevated in HER2+ cancers and correlates with resistance to anti-HER2 therapy and low relapse free survival (RFS) rates, post-treatment. The NEDD9 induces higher rates and earlier formation of preneoplastic, benign lesions, such as mammary intraepithelial neoplasia (MIN) and DCIS. In our study, we found that overexpression of NEDD9 correlates with the expansion of luminal cells and a significant increase in the luminal progenitors, characterized by dual K5+/K8+ staining. The increase in proliferation correlates with the increased activity of MAPK and AURKA proliferation pathways. Thus, depletion of the NEDD9 protein might provide significant benefits in treating therapy resistant HER2+ breast cancers. The expression of NEDD9 might serve as a prognostic marker for therapy response.

## Figures and Tables

**Figure 1 cancers-15-01119-f001:**
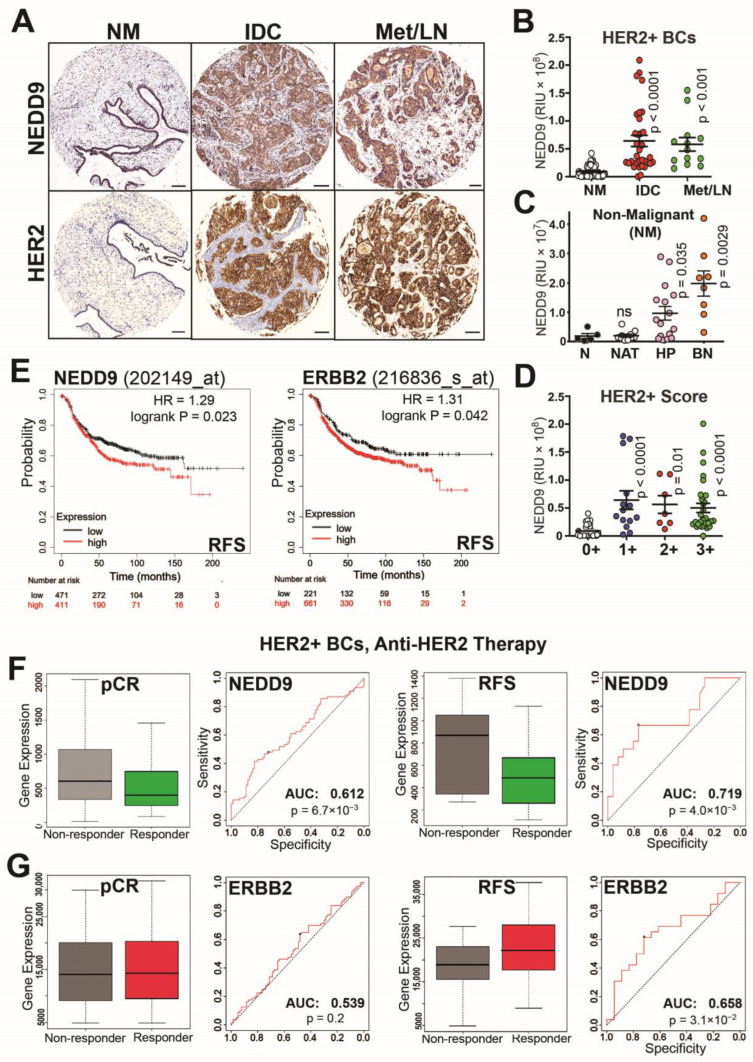
NEDD9 is overexpressed in HER2+ breast cancers and correlates with disease progression, poor survival, and anti-HER2 therapy response. (**A**) Representative images of immunohistochemical staining of tissue microarray (TMA, n = 90, including 40 nonmalignant (NM): 6 normal, 10 normal adjacent to tumor tissue-NAT, 16 hyperplasia, and 8 benign) with anti-NEDD9 and -HER2 antibodies, developed with DAB (brown) and hematoxylin-nuclei (blue); HER2+ invasive ductal carcinoma (IDC, n = 36), metastatic invasive ductal carcinoma-lymph nodes (Mets-LN, n = 14). Scale bar 300 mm. The de-identified tumor biopsy information is shown in Appendix A. (**B**) Quantification of the intensity of DAB staining and positivity (percentage of cells stained positively in the same core stained by anti-HER2 or -NEDD9) by automated AperioScope imaging tool as in (**A**); patient stratified based on diagnosis (**B**) NM-40, IDC-36, Mets/LN-14. (**C**) Nonmalignant cases were further stratified: 16 normal/NAT, 16 hyperplasia, 8 benign; or (**D**) based on HER2+ pathological score: NM 40 (0+), 1+(14), 2+(6), 3+(30). One-way ANOVA with Dunnett’s multiple comparisons test, *p*-values as indicated in figure panel. (**D**) Kaplan–Meier patient survival plots were generated by the Kaplan–Meier Plotter (http://www.kmplot.com, accessed on 20 December 2022) to determine the effect of NEDD9 or HER2 expression (microarray data Affy ID: 202149_at) on the relapse free survival (RFS) of 882 patients with HER2+ (array) breast cancer. Differences between expression levels shown by hazard ratio and *p*-values are provided in each Kaplan-Meier plot. Hazard ratios and *p*-value as indicated in figure panel. (**E**) pROC assessment of anti-HER2 therapy response and NEDD9 expression levels in HER2+ breast cancer patients (HER2-array). (**F**) Input settings for response—pathological complete response, HER2 status—positive, Treatment—anti-HER2 Therapy. Responders n = 80, Non-responders = 77. AUC = 0.612, *p* = 6.7 × 10^−3^. pROC assessment of anti-HER2 therapy response and NEDD9 expression levels in HER2+ breast cancer patients (HER2-array). Input settings for response—relapse-free survival at 5 years, HER2 status—positive, Treatment—anti-HER2 Therapy. Responders n = 26, Non-responders = 18. AUC = 0.719, *p* = 4.0 × 10^−3^. pROC assessment of anti-HER2 therapy response and Erbb2 expression levels in HER2+ breast cancer patients (HER2-array). (**G**) Input settings for response –pathological complete response, HER2 status—positive, Treatment—anti-HER2 Therapy. Responders N = 80, Non-responders = 77. AUC = 0.539, *p* = 0.2. pROC assessment of anti-HER2 therapy response and Erbb2 expression levels in HER2+ breast cancer patients (HER2-array). Input settings for response—relapse-free survival at 5 years, HER2 status—positive, Treatment—anti-HER2 Therapy. Responders n= 26, Non-responders = 18. AUC = 0.658, *p* = 3.1 × 10^−2^.

**Figure 2 cancers-15-01119-f002:**
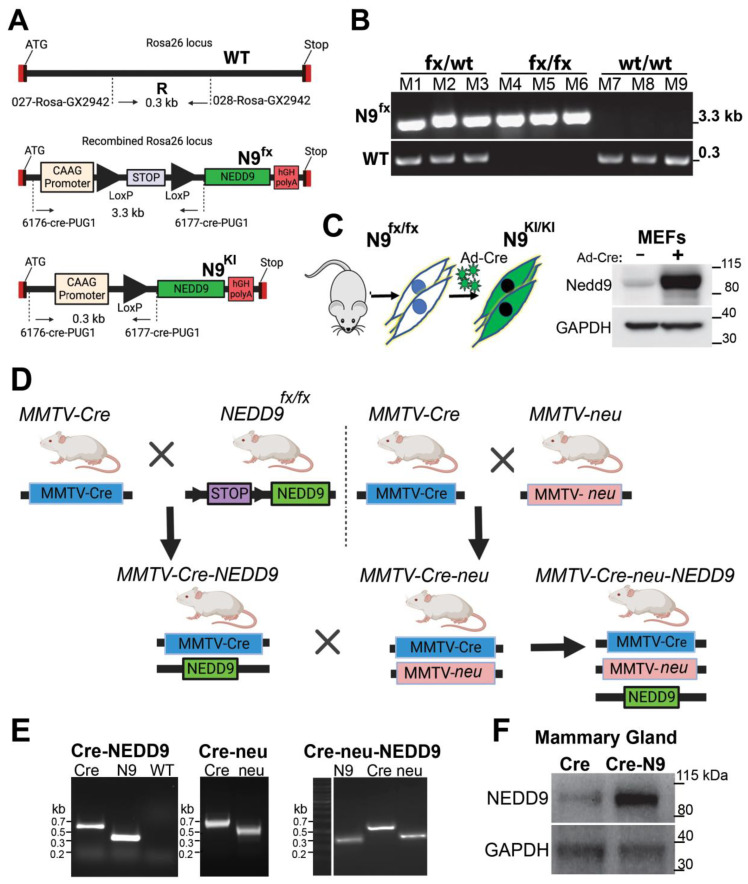
Design and expression of Cre inducible NEDD9 knock-in mouse model. (**A**) Design of Cre recombinase-inducible NEDD9 cDNA knock-in vector for homologous recombination at the Rosa26 locus (top). In the presence of Cre recombinase, the stop cassette is resected out of the genome due to flanking LoxP sites (middle), resulting in CAAG promoter-driven NEDD9 expression (bottom). (**B**) Genotyping results in triplicate, indicating the presence of either the heterozygote (fx/wt) (M1-3), homozygous (fx/fx) (M4-6), or wild type Rosa26 loci (wt/wt) (M7-9). (**C**) Schematic outline of adenovirus Cre delivery into mouse embryonic fibroblasts (MEFs), isolated from NEDD9fx/fx mice (n = 3). MEFs were incubated with GFP (−) or adenoviral GFP-Cre (+) virus particles. Western blot analysis of MEF’s lysates with anti-NEDD9. -GAPDH is a loading control. (**D**) Mating scheme of mice used in study. (**E**) Representative genotyping results of mice (genotypes as indicated in (**D**)), using genomic DNA and target-specific primers, showing presences of Rosa 26 loci (WT), NEDD9 (N9), Cre Recombinase (Cre), Erbb2 (neu). (**F**) Western blot analysis of lysates prepared from the mammary gland tissue of MMTV-Cre-control or MMTV-Cre-NEDD9 mice (as in (**E**), n = 3) with anti-NEDD9. -GAPDH is a loading control.

**Figure 3 cancers-15-01119-f003:**
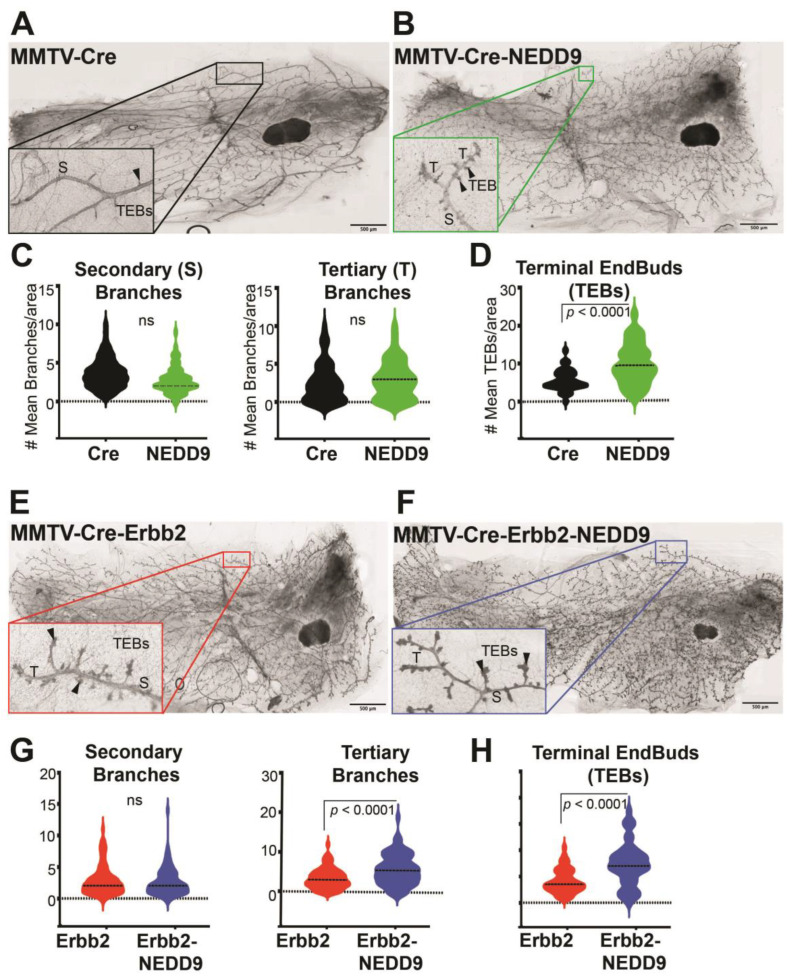
NEDD9 overexpression alters mammary gland architecture. (**A**,**B**) Representative images of mammary gland whole mounts prepared from MMTV-Cre and MMTV-Cre-NEDD9. Scale bar—500 mm; Inset—400×; S—Secondary Branch; T—Tertiary Branch; TEB—Terminal End Bud. (**C**) Quantification of secondary and tertiary branches (fields of view= 5, n = 5 per genotype). (**D**) Quantification of terminal end buds (TEBs) (fields of view= 5, n = 5 per genotype). Non-parametric Student’s *t*-test, ±S.E.M; ns—not significant; TEBs—*p* < 0.0001. (**E**,**F**) Representative images of mammary gland whole mounts prepared from MMTV-Cre-Erbb2 and MMTV-Cre-Erbb2-NEDD9 mice. Scale bar—500 μm; Inset—400×; S—Secondary Branch; T—Tertiary Branch; TEB—Terminal End Bud. (**G**) Quantification of secondary and tertiary branches and (**H**) terminal end buds (TEBs) (fields of view= 5, n = 5 per genotype). Non-parametric Student’s *t*-test, ±S.E.M; ns- not significant; *p* < 0.0001 for tertiary branches and TEBs.

**Figure 4 cancers-15-01119-f004:**
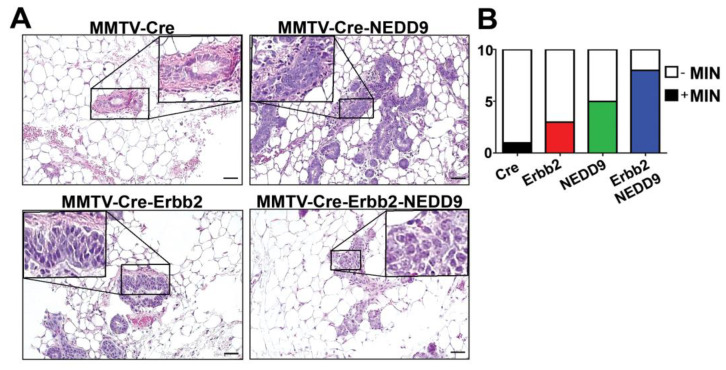
NEDD9 overexpression is associated with mammary intra-epithelia neoplasia. (**A**) Representative images of mouse mammary glands were stained with hematoxylin and eosin (H&E), genotypes as indicated in the figure. Scale bar—20 mm; Insets—200×. A total of 40 slides from 16-week-old nulliparous female mice with four genotypes were quantified. The average area per slide was 100–120 mm^2^. MIN was determined by lesions, characterized by glands, and lined by 1–2 layers of atypical epithelial cells with enlarged nuclei, clumped chromatin, variable nuclear size and shape, and an increased mitotic rate. (**B**) Quantification of mice with MIN per group (n = 10/genotype). White—no MIN detected, MIN+ positive (colored). Fisher’s exact test, *p* = 0.0137, number reported in Table 1.

**Figure 5 cancers-15-01119-f005:**
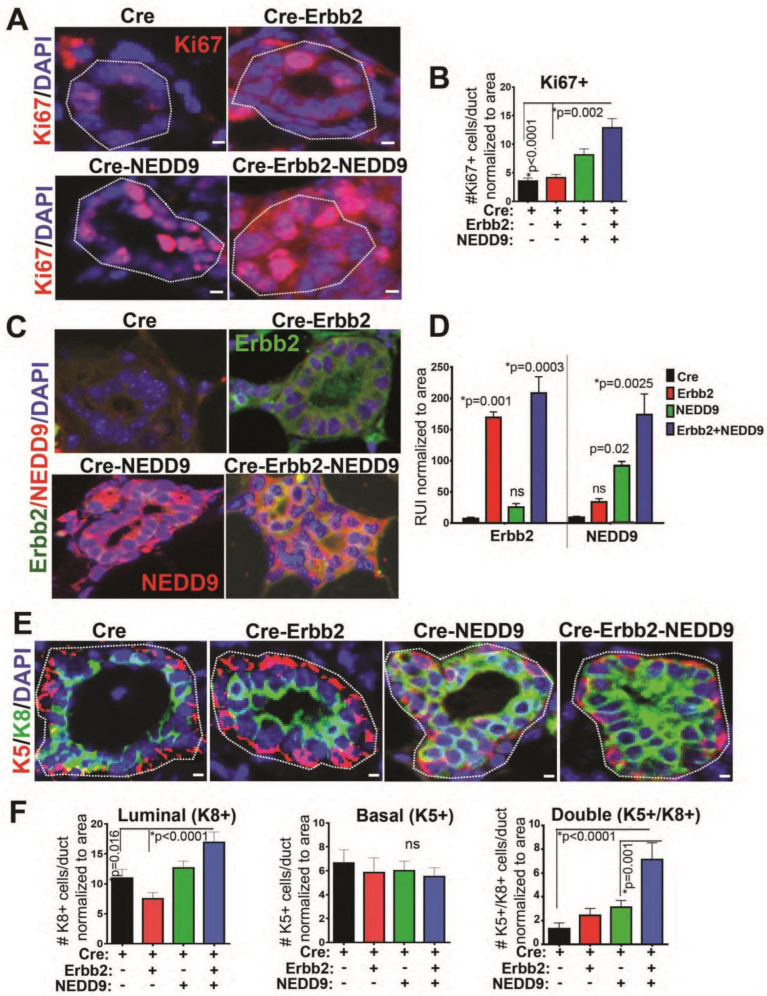
NEDD9 overexpression increases proliferation of luminal cells. (**A**) Representative mammary gland images (3D confocal projections, cropped from whole mammary gland scanned) from n = 3–6 mice per genotypes, as indicated in the figure. Fluorescent-IHC staining with anti-Ki67 (red) and DAPI—nuclei (blue). Scale bar—5 mm. (**B**) Quantification of number of positive cells/duct normalized to area (n = 5 ducts/mouse, n = 5 per genotype). Data presented as mean ± S.E.M. *p*-value, indicated in figure panels. One-way ANOVA, Tukey’s multiple comparisons: Cre vs. Cre-Erbb2 is non-significant (ns); Cre vs. Cre-NEDD9 is *p* = 0.009; Cre vs. Cre-Erbb2-NEDD9 is * *p* < 0.0001; Cre-Erbb2 vs. Cre-NEDD9 is * *p* = 0.0002, or vs. Cre-Erbb2-NEDD9 is * *p* < 0.0001; Cre-NEDD9 vs. Cre-Erbb2-NEDD9 is * *p* = 0.05. (**C**) Representative mammary gland images. Fluorescent-IHC staining with anti-Erbb2 (green), anti-NEDD9 (red), and DAPI—nuclei (blue). Scale bar—5 mm. (**D**) Quantification of median relative units of intensity of Erbb2 (left) and NEDD9 (right), as in panel (**C**), normalized to area (n = 5 ducts/mouse, n = 3 per genotype). Data presented as mean ± S.E.M. *p*-value indicated in figure panels. One-way ANOVA, Tukey’s multiple comparisons of Erbb2 expression (left): Cre vs. Cre-NEDD9 is non-significant (ns); Cre vs. Cre-Erbb2 is * *p* = 0.001; Cre vs. Cre-Erbb2-NEDD9 is *p* = 0.0003. One-way ANOVA, Tukey’s multiple comparisons of NEDD9 expression (right): Cre vs. Cre-Erbb2 is ns; Cre vs. Cre-NEDD9 is * *p* = 0.02; Cre vs. Cre-Erbb2-NEDD9 is *p* = 0.0025. (**E**) Representative mammary gland images (3D confocal projections, cropped from whole mammary gland scanned) from n = 3–6 mice per genotypes, as indicated in the figure. Fluorescent-IHC staining with anti-cytokeratin 8 (green), or cytokeratin 5 (red), and DAPI—nuclei (blue). Scale bar—5 mm. (**F**) Quantification of the number of luminal (K8+), basal (K5+), and double positive (K5+/K8+) cells/duct normalized to area/mammary gland ducts (n = 5 ducts/mouse, n = 5 per genotype). One-way ANOVA, Tukey’s multiple comparisons ±S.E.M. *p*-value indicated in figure panels. ns—not significant. Cre vs. Cre-Erbb2-NEDD9 is * *p* = 0.01; Cre-NEDD9 vs. Cre-Erbb2 is * *p* = 0.02, or vs. Cre-Erbb2 is *p* < 0.0001; K5+/K8+ Cre vs. Cre-Erbb2-NEDD9 is *p* < 0.0001; Cre-NEDD9 vs. Cre-Erbb2-NEDD9 is *p* = 0.001.

**Figure 6 cancers-15-01119-f006:**
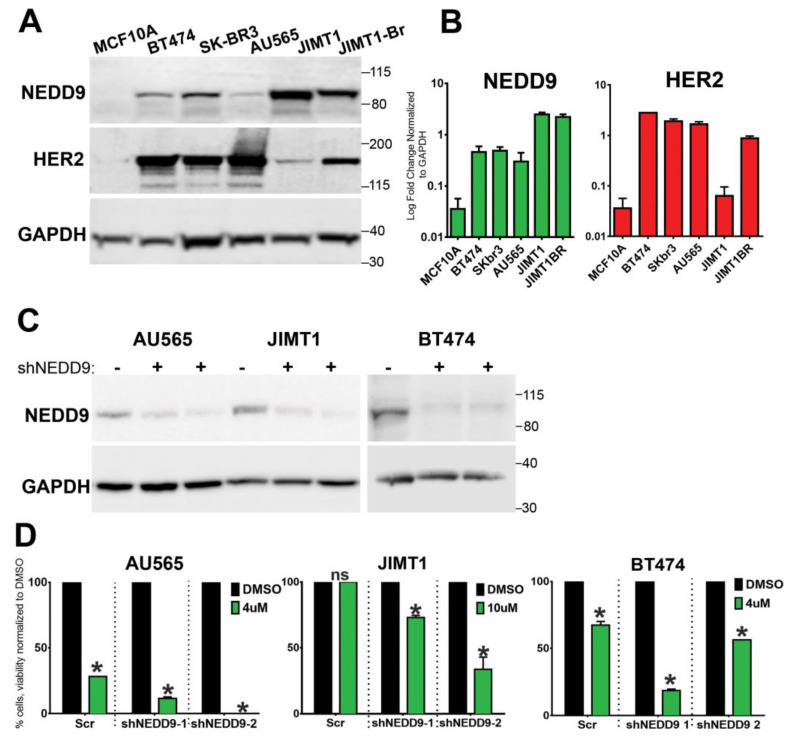
NEDD9 is overexpressed in human HER2+ breast cells and supports resistance to lapatinib. (**A**) Western blot (WB) analysis of MCF10A, BT-474, SK-BR-3, AU565, JIMT-1, and JIMT-1-Br3 (brain metastases subline of JIMT1), using anti-NEDD9, -HER2, and -GAPDH as loading controls. (**B**) WB-based quantification of NEDD9 (green) and HER2 (red) protein expression in cell lines as in (**A**). n = 3 independent experiments. Parson correlation coefficient r = 0.972, with exception of JIMT1, where expression of NEDD9 and HER2 do not correlate. (**C**) Western blot analysis of lysates of AU565, JIMT1, and BT474 cells treated with scramble (−) non-targeting control or two different shRNAs against NEDD9 (shNEDD9), using anti-NEDD9 and -GAPDH as loading controls. (**D**) Cell viability assessment using Cyquant dye with different concentrations of lapatinib, as indicated in the figure. Two-way ANOVA, Tukey’s multiple comparison +/− SEM. AU565, JIMT1, BT474, * *p* < 0.0001 (DMSO vs. shNEDD-1 or -2), ns—non significant. The original western blots of Figure 6 is Appendix A.

**Figure 7 cancers-15-01119-f007:**
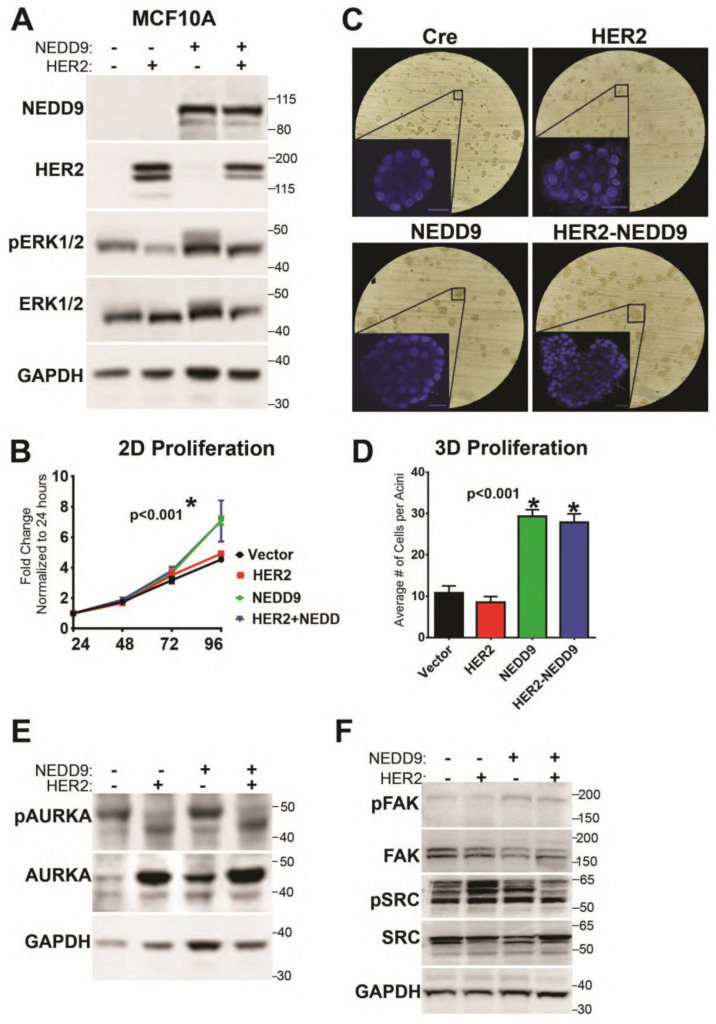
NEDD9 expression increases proliferation through alteration of Aurora A Kinase expression. (**A**) Western blot analysis of MCF10A cells overexpressing NEDD9, HER2, or both with anti-NEDD9, -HER2, -pERK1/2, -ERK1/2, and -GAPDH as a loading control. n = 3 independent experiments. (**B**) MTT proliferation assay using cells as in (**A**), measured at 24, 48, 72, and 96 h. Two-way ANOVA +/− SEM, Tukey’s multiple comparisons: vector vs. NEDD9 or HER2+NEDD9—* *p* < 0.001 at 96 h; vector vs. HER2 is ns. (**C**) Representative brightfield images of mammary acini using cells as in (**A**), inset nuclei (blue)—DAPI (3D confocal projections every 10 mm, total of 200 mm). Scale bar—20 mm. (**D**) Quantification of number of cells per acini using confocal imaging (n = 50 acini total/cell line) in 3 independent experiments. One-way ANOVA +/− SEM, Tukey’s multiple pairwise comparisons: vector vs. NEDD9 or HER2+NEDD9—* *p* < 0.001; HER2 vs NEDD9 or HER2+NEDD9—* *p* < 0.001. (**E**) Western blot analysis of MCF10A cells as in (−), with anti- pAURKA, -AURKA, -GAPDH as loading controls. (**F**) Western blot analysis of MCF10A cells with anti-pFAK, -FAK, -pSrc, -Src, and GAPDH as loading controls. The original western blots of Figure 7 is Appendix A.

**Table 1 cancers-15-01119-t001:** Analysis of Mammary Intraepithelial Neoplasia (MIN) in mice.

Genotype	MIN+, N (%)	MIN−, N(%)
MMTV-Cre	1 (10%)	9 (90%)
MMTV-Cre-Erbb2	3 (30%)	7 (70%)
MMTV-Cre-NEDD9	5 (50%)	5 (50%)
MMTV-Cre-Erbb2-NEDD9	8 (80%)	2 (20%)

MIN+ mammary intraepithelial neoplasia positive or—negative mice. N = 10 per each genotype. Fisher’s Exact test statistics, *p* = 0.0137.

## Data Availability

Publicly available datasets were analyzed in this study. Data can be found at: https://www.cbioportal.org (accessed on 20 December 2022), https://kmplot.com/analysis/index.php?p=service&cancer=breast (accessed on 20 December 2022), and https://www.rocplot.org/site/treatment (accessed on 20 December 2022).

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
