# Peer review of "NEDD9 Overexpression Causes Hyperproliferation of Luminal Cells and Cooperates with HER2 Oncogene in Tumor Initiation: A Novel Prognostic Marker in Breast Cancer"

_cancers, 2023, doi:10.3390/cancers15041119_

Round 1
Reviewer 1 Report
In this report, Purazo et al., studied the impact of NEDD9 overexpression on HER2 breast cancers. To this aim they performed functional studies in human cell lines and mouse models and corelative studies using human samples and public databases.
The mouse models provide convincing evidences about the priming role of NEDD9 on mammary gland tumorigenesis, since overexpression of NEDD9 in the epithelium is associated with early lesions. Moreover, in the ERBB2 overexpression background NEDD9 overexpression further increase the number of preneoplastic lesions. Functional exploration showed an increase of proliferation.
The authors studied the impact of NEDD9 expression on human tissues and cell lines. By TMA analysis and data mining they concluded that NEDD9 is overexpressed in HER2-positive BC and its expression is correlated with disease progression. In addition, high NEDD9 mRNA levels correlates with shorter Relapse Free Survival in HER2 + tumor patients and importantly is a predictive marker of pathological complete response towards Trastuzumab, if I am correct. Then the authors studied functionally NEDD9 expression in human breast-derived cell lines. They showed overexpression of NEDD9 protein in HER2-positive cell lines. Gain of function in MCF10A - a non-cancerous breast cancer cell line - shows increase proliferation in 2D and 3D, interestingly no additive effect was observed in the presence of overexpressed HER2. Mechanistic exploration pointed to the activation of the Aurora A kinase pathway. Loss of Function in HER2+ cell line showed that NEED9 contributes to lapatinib resistance, suggesting that targeting NEDD9 could be of interest for the treatment of HER2+ patients whose tumors are not sensitive to conventional anti-HER2 treatments.
This article reports interesting observations and findings but it contains flaws that should be addressed.
Major concerns
1 two paragraphs are duplicated page 3, paragraph 2-1 is similar to 2.2 and page 5, 2-10 is identical to 2-11
2 - The TMA analysis is puzzling, and I am worried about the conclusions. Are 115 samples tested in the TMA supposed to be representative of breast cancers? The ratio of HER2 positive cases is way above what is expected (~50%, Table S1). Is there a bias in recruitment or are these selected samples? The rational and the method of selection should be explained. NAT is defined as normal/non tumoral adjacent tissue, but my impression from Figure 1B is that NAT (not defined in the legend) includes early lesions? In addition, two samples called NAT or AT have an HER2 histoscore of 3+ (table S1) is it true? The conclusion of Figure 1B claims a correlation between NEDD9 expression and disease progression p 8 lanes 392 – 303 and page 9 lanes 394. But the few begin lesions are not plotted and the mean intensities for NEDD9 expression are similar between the IDC and the Met/LN groups. TI would rather conclude that NEDD9 is overexpressed in malignant samples (IDC and Met/LN) compared to normal tissues? Figure 1C is also confusing are all the samples plotted including normal samples? I do see a negative correlation: tumor with a high histoscore (3+) have a mean intensity lower than the other tumor samples. I am not sure about the meaning of the conclusion p9 lanes 395-396. The legend of Figure 1 is not clear, NAT should be better defined, what is the meaning of NAT-50 and then NAT-56 ?
3 - regarding the data mining. KM plotter, the authors chose to present only the HER2+ subtype results, what about the other subtypes? This should be done and commented. It might reinforce the link with HER2. Rocplot, this analysis shows that NEDD9 could be a predictive maker for anti-HER2 therapy. This is an important finding, there is a lack of predictive marker and more patients are treated with chemotherapy and anti-HER2 in a neo-adjuvant setting. I found that Fig 1D was not optimally explained, were those patients treated in a neoadjuvant or adjuvant setting and only with trastuzumab?
4 – The description of the mouse model and phenotype were convincing and well written; however, I wonder if the author studied tumor burden at later stages to explore the role of NEDD9 on tumor progression and metastasis. They should comment on that point. The authors did not study apoptosis?
5- The expression of NEDD9 was studied in human HER2+ cell lines what about the other subtypes? AKT signaling was not tested or it is unchanged? Looking rapidly at the CCLE data base, it seems that NEED9 overexpression is not specific of HER2+ cell lines.
Minor concerns
Figure 2 is called before Fig1, there are typos that should be corrected.
The acini preparation method should be described, reference 39 is a very nice paper but it does not specifically describe acini formation.
Author Response
MS# Cancers-2150513. Revision.
Date: January 19th 2023
NEDD9 overexpression causes hyperproliferation of luminal cells and cooperates with HER2 oncogene in tumor initiation: a novel prognostic marker in breast cancer.
Author's Response to Reviewer's Critique:
We appreciate the reviewers and editors for their thorough review of our manuscript and for providing constructive feedback on its critical findings. The text has undergone scientific and English editing to improve the reading and presentation of our work. The changes are easy to track in the text to streamline the review process. Below are point-by-point responses to the raised concerns. We hope the revised manuscript addresses all the points raised and will be acceptable for publication.
Reviewer #1 (Comments to the Author):
Reviewer #1: 1 - two paragraphs are duplicated page 3, paragraph 2-1 is similar to 2.2 and page 5, 2-10 is identical to 2-11
Authors: We apologize for missing this and have removed both duplicate paragraphs under sections 2.2 & 2.11.
Reviewer #1: 2 - The TMA analysis is puzzling, and I am worried about the conclusions.
Authors: We appreciate the Reviewer's comments on the TMA analysis that allowed us to improve data presentation and strengthen our conclusions. We hope detailed answers and clarification of the sample IDs below will resolve the raised concerns.
- Reviewer #1: Are 115 samples tested in the TMA supposed to be representative of breast cancers?
Authors: "The samples in the TMA were selected to represent two groups: 1. non-malignant (NM) biopsies that include normal tissue from healthy donors, normal adjacent to the tumor tissue (NAT), hyperplasia, and benign lesions; 2. Malignant HER2+ biopsies included carcinomas, invasive ductal carcinomas, and lymph node metastases." This text was added to section 2.10 of the revised manuscript.
- Reviewer #1: The ratio of HER2 positive cases is way above what is expected (~50%, Table S1). Is there a bias in recruitment, or are these selected samples? The rational and the method of selection should be explained.
Authors: The rational and selection methods have been added to the text. These are selected samples as described above. The rationale was to evaluate NEDD9 expression in HER2+ breast cancers (BCs) with variable HER2 expression (score) and compare it with non-malignant stages and metastatic lesions. It is well established that NEDD9 expression correlates with the onset of EMT and increased migration/invasion and metastasis. Our group has previously reported the expression of NEDD9 in triple-negative breast cancers (TNBCs). Other groups have reported on the NEDD9 in ER+ breast cancers. While little is known about the role, NEDD9 plays in HER2+ BCs and the early stages of tumorigenesis. The analysis of NEDD9 expression across non-malignant groups shows a significant increase in NEDD9 levels as early as hyperplasia and benign lesions supporting our conclusions. These results, representative images, and quantification were added to Figure 1 and Supplementary Figure 1A.
- Reviewer #1: NAT is defined as normal/non-tumoral adjacent tissue, but my impression from Figure 1B is that NAT (not defined in the legend) includes early lesions? In addition, two samples called NAT or AT have an HER2 histoscore of 3+ (table S1) is it true?
Authors: We appreciate Reviewer's notes on this and acknowledge that NAT in the original submission included all non-malignant samples. NAT is an abbreviation for normal tumor-adjacent tissue, as Reviewer pointed out correctly. We defined this group as non-malignant tissue (NM) in the revised manuscript. The samples included in this group are now correctly identified. Supplementary Table 1 was revised to reflect these changes. We also removed 16 samples corresponding to various inflammatory breast pathologies due to high levels of NEDD9 expression in immune cells, specifically lymphocytes. The NEDD9 (also called Cas-L) was discovered in lymphatic tissue and is critical for lymphocyte maturation. The inflammation biopsies have significant infiltration of immune cells, complicating the analysis. The graphs in Figure 1B-D were modified to reflect this change. In addition, the NAT and AT samples mentioned above have been corrected based on the pathology records. In total, 90 samples were analyzed, including 6 normal breast tissue, 10 normal adjacent tumor tissue, 16 hyperplasia lesions, 8 benign lesions, 36 malignant tumor samples, and 14 lymph node metastases. The information in material and methods was revised accordingly.
- Reviewer #1: The conclusion of Figure 1B claims a correlation between NEDD9 expression and disease progression p 8 lanes 392 – 303 and page 9 lanes 394. But the few begin lesions are not plotted and the mean intensities for NEDD9 expression are similar between the IDC and the Met/LN groups. TI would rather conclude that NEDD9 is overexpressed in malignant samples (IDC and Met/LN) compared to normal tissues?
Authors: We agree with Reviewer's comment. In the revised manuscript, the non-malignant biopsies (normal, NAT, Hyperplasia, and Benign) were plotted separately to evaluate the correlation between the progression of early stages of the disease and NEDD9 expression. Both hyperplasia and benign samples show a statistically significant difference compared to normal breast tissue or NAT. These new findings are included in Figure 1C. The representative images of IHC stating were included in Supplementary Figure 1A.
- Reviewer #1: Figure 1C is also confusing are all the samples plotted including normal samples? I do see a negative correlation: tumor with a high histoscore (3+) have a mean intensity lower than the other tumor samples. I am not sure about the meaning of the conclusion p9 lanes 395-396. The legend of Figure 1 is not clear, NAT should be better defined, what is the meaning of NAT-50 and then NAT-56 ?
Authors: The pathologist scored the non-malignant samples, including normal and benign groups, as 0 for HER2 expression. The difference in mean intensity between 1+, 2+, or 3+ is non-significant, but all three are significantly different from the non-malignant group. The conclusion on p9 has been modified accordingly. The total of 40 samples since we excluded the inflammation group. The NAT corresponds to 10 samples in total. The groups were identified correctly in the revised manuscript. We apologize for the lack of clarity in the original manuscript.
Reviewer #1: 3 - regarding the data mining. KM plotter, the authors chose to present only the HER2+ subtype results, what about the other subtypes? This should be done and commented. It might reinforce the link with HER2.
Authors: To address this concern, we have added two additional panels in Supplementary Figure 1B and discussed the results in section 3.1. The NEDD9 expression using KM plotter datasets has been analyzed in estrogen receptor-positive (ER+) breast cancers and basal breast cancers. We have previously published two studies showing upregulation of NEDD9 in breast cancers, especially in triple-negative BCs. In ER+ subtype, the upregulation of NEDD9 mRNA correlates with better outcomes, indicating the different roles of NEDD9 in ER+ BCs compared to HER2+ cancers.
Reviewer #1: Rocplot, this analysis shows that NEDD9 could be a predictive maker for anti-HER2 therapy. This is an important finding, there is a lack of predictive marker and more patients are treated with chemotherapy and anti-HER2 in a neo-adjuvant setting. I found that Fig 1D was not optimally explained, were those patients treated in a neoadjuvant or adjuvant setting and only with trastuzumab?
Authors: We agree with Reviewer's notes, but unfortunately, the ROCplot data sets have not been outlined in sufficient depth to address this point. We thoroughly reviewed the reference provided in the original report that outlines the ROC plotter tool. According to that publication, most patients in the pCR and RFS cohorts have received chemotherapy. It is not clear if the settings were neoadjuvant or adjuvant. The small patient cohort used in our analysis was treated with lapatinib (n= 65) and trastuzumab (n= 186). Patients included in RFS ROC plots received anty-HER2 therapy in adjuvant settings. This information has been added to section 2.18. We run additional analyses and show that NEDD9 expression is highly predictive, specifically in HER2+ cases treated with anti-HER2-therapy or FEC chemotherapy (Fluorouracil, Epirubicin, Cyclophosphamide; AUC=0.75~0.8, p=6.2x10-11), but not predictive when patients were treated with FAC(Fluorouracil, Adriamycin, Cytoxan, AUC=0.5, p=0.49) or Antrocyclines (AUC=0.507, p=6x10-3). The NEDD9 expression does not influence RFS (AUC=0.51, p=0.35) or pCR response to aromatase inhibitors or Tamoxifen in ER+ breast cancers (AUC=0.505, p=0.47), indicative of unique role NEDD9 plays in HER2+ cancers. The results have been included in Supplementary Figure 2 and discussed in section 3.1.[1]
Reviewer #1: 4 – The description of the mouse model and phenotype were convincing and well written; however, I wonder if the author studied tumor burden at later stages to explore the role of NEDD9 on tumor progression and metastasis. They should comment on that point. The authors did not study apoptosis?
Authors: This is a great point, and we are finalizing a separate manuscript that focuses on the later stages of HER2 tumorigenesis in the models we made. In fairness to the team that completed that study, we kindly ask not to introduce that data here. The other manuscript has in-depth metastasis and tumor burden analysis along with signaling and mechanisms. We hope it will be submitted shortly, and the findings will be shared. We did not study apoptosis in this study since we have not observed cell death in the mammary glands or in vitro models. The cell death analysis has been done for the later stages, where we show that NEDD9 increases incidence and tumor growth supporting our current findings.
Reviewer #1: 5- The expression of NEDD9 was studied in human HER2+ cell lines what about the other subtypes? Looking rapidly at the CCLE data base, it seems that NEED9 overexpression is not specific of HER2+ cell lines.
Authors: We agree with Reviewer that NEDD9 overexpression was observed in other subtypes of BCs. Other groups and we have published reports documenting these observations. It was shown in Triple Negative Breast cancers and in ER+ breast cancer [2, 3] These references have been incorporated in the discussion section. The novelty of our study is that NEDD9 expression and its impact on treatment outcomes in HER2+ BCs was not previously documented, and the mechanisms of its action were defined.
Reviewer #1: 6- AKT signaling was not tested or it is unchanged?
Authors: The AKT signaling was not tested in this study.
Minor concerns
Reviewer #1: Figure 2 is called before Fig1, there are typos that should be corrected.
Authors: We corrected the order of the figures and typos.
Reviewer #1: The acini preparation method should be described, reference 39 is a very nice paper but it does not specifically describe acini formation.
Authors: To address this concern, we have added PMID: 12798140 as part of the appropriate methods section. This additional reference is a method paper by the same group that describes the methodology in depth.
1. Fekete, J.T. and B. Győrffy, ROCplot.org: Validating predictive biomarkers of chemotherapy/hormonal therapy/anti-HER2 therapy using transcriptomic data of 3,104 breast cancer patients. International Journal of Cancer, 2019. 145(11): p. 3140-3151.
- McLaughlin, S.L., et al., NEDD9 Depletion Leads to MMP14 Inactivation by TIMP2 and Prevents Invasion and Metastasis. Molecular Cancer Research, 2014. 12(1): p. 69-81.
- Law Susan, F., et al., The Docking Protein HEF1 Is an Apoptotic Mediator at Focal Adhesion Sites. Molecular and Cellular Biology, 2000. 20(14): p. 5184-5195.
Reviewer 2 Report
The article investigates the role of NEDD9 overexpression in breast cancer initiation using GEMM and cell lines. The role of NEDD9 in tumor initiation has been studied in MMTV-HER2/ERBB2/neu mouse mammary tumor model (PMID: 23318423). While this manuscript uses NEDD9 overexpression as the experimental approach, the results lead to similar conclusions. Unfortunately, it does not go beyond what has already been shown. Overall, the manuscript is well-written, and the figures are well-presented. Here are some comments that can help improve the manuscript:
1- General comment:
Generating cell lines from the established mouse models would provide additional mechanistic insight into the NEDD9/Her2 crosstalk.
1- In Fig 5A, the authors evaluate the level of proliferation markers (ki67) without characterizing Her2 and NEDD9 at the cellular level. As western blot does not provide any details on cellular heterogeneity, using immunofluorescence imaging to depict direct correlations between Nedd9, Her2, and ki67 in these models would be appropriate.
2- While the number of basal cells is not influenced as shown in Fig 5C, the images suggest an increased proportion of cells with dual expression of Keratin 5 and 8. This requires further evaluation. It would be important to also ascertain whether these K8+/K5+ cells associate with NEDD9 expression at the cellular level.
3- Fig 6A shows increased NEDD9 expression in Her2+ breast cancer models with the only control being MCF10A. There is still a possibility that other types of cancer including TNBCs show similar expression of NEDD9. The specificity of NEDD9 to the Her2+ subtype needs to be further validated. Publicly available datasets could be used to support NEDD2 distribution in various breast cancer subtypes.
4- The positive correlation between NEDD2 and Her2 needs further evaluation.
5- From Fig 7 it is clear that NEDD9 does not require Her2 to activate the MAPK signaling and promote cell proliferation. Therefore, the association of NEDD9 to Her2 signaling does not seem relevant especially when the authors still attempt to block Her2 in Fig 6D. Considering that Her2 required NEDD9 to activate its downstream signaling, the focus of this manuscript should have been on targeting NEDD9.
6- In line with the above, the authors must test pharmacological drugs targeting NEDD9 (if available) in Her2+ breast cancer models that are sensitive and resistant to anti-her2 treatment.
Author Response
MS# Cancers-2150513. Revision.
Date: January 19th 2023
NEDD9 overexpression causes hyperproliferation of luminal cells and cooperates with HER2 oncogene in tumor initiation: a novel prognostic marker in breast cancer.
Author's Response to Reviewer's Critique:
We appreciate the reviewers and editors for their thorough review of our manuscript and for providing constructive feedback on its critical findings. The text has undergone scientific and English editing to improve the reading and presentation of our work. The changes are easy to track in the text to streamline the review process. Below are point-by-point responses to the raised concerns. We hope the revised manuscript addresses all the points raised and will be acceptable for publication.
Reviewer #2 (Comments to the Author):
- Reviewer #2: Generating cell lines from established mouse models would provide additional mechanistic insight into the NEDD9/Her2 cross talk.
Authors: We agree with Reviewer that generating cell lines from established mouse models would provide additional mechanistic insight, and we have been able to make cell lines from mouse tumors at a later stage. These cell lines are a foundation for our separate manuscript focusing on later stages and metastasis under preparation. Unfortunately, those cell lines are not representative of early tumorigenesis. Biological materials from early hyperplastic/benign lesions are limited in quantity and have prevented us from producing established cell lines. We have attempted to grow dissociated mammary gland cells from the early stages but were unsuccessful. We used the immortalized mammary gland human cell line MCF10A, which was previously well-characterized and used by the scientific community to model early tumorigenesis.
- Reviewer #2: In Fig 5A, the authors evaluate the level of proliferation markers (ki67) without characterizing Her2 and NEDD9 at the cellular level. As western blot does not provide any details on cellular heterogeneity, using immunofluorescence imaging to depict direct correlations between Nedd9, Her2, and ki67 in these models would be appropriate.
Authors: We appreciate the Reviewer for bringing this to our attention and agree that it will strengthen the correlation. To address this concern, we have conducted fluorescent IHC using anti-NEDD9 and -HER2 antibodies. We have added images to Figure 5C-D.
- Reviewer #2: While the number of basal cells is not influenced as shown in Fig 5C, the images suggest an increased proportion of cells with dual expression of Keratin 5 and 8. This requires further evaluation. It would be important to also ascertain whether these K8+/K5+ cells associate with NEDD9 expression at the cellular level.
Authors: We agree with Reviewer and appreciate this comment, which led to exciting findings we missed in the original manuscript. We have quantified the number of dual positive cells and confirmed Reviewer's observation. The additional graph was added to Figure 5F. Unfortunately, in the time provided for review (10 days), we could not complete the mammary gland staining with anti-NEDD9 antibodies due to technical problems associated with the same species staining protocol. The following text was added to the Discussion to strengthen the conclusions. "The normal breast tissue contains subpopulations of mammary stem cells (MaSC) and progenitors. The origin of different BC tumor subtypes often correlates with one of the subgroups. The basal-like and HER2+ BCs resemble the luminal progenitor cells. These cells are often dual-positive KRT5+/KRT8+ and found in the luminal layer of the duct[4]. In agreement with our findings in NEDD9 transgenic mice, the HER2-enriched cell lines BT474 and SKBR3 showed expression of both keratin 5 and 8 [4]. The increase in double-positive luminal cells supports the conclusion that NEDD9 overexpression increases the pool of cells considered a cell of origin for transformation in the HER2+ subtype."
- Reviewer #2: Fig 6A shows increased NEDD9 expression in Her2+ breast cancer models with the only control being MCF10A. There is still a possibility that other types of cancer including TNBCs show similar expression of NEDD9. The specificity of NEDD9 to the Her2+ subtype needs to be further validated. Publicly available datasets could be used to support NEDD2 distribution in various breast cancer subtypes.
Authors: We wanted to stress that NEDD9 overexpression was previously documented in other subtypes. We provided multiple references from our group to support this notion. In the revised manuscript, we clarify that our focus on NEDD9 in HER2+ BCs does not exclude its role in other BCs; it brings into focus its correlation between both protein and mRNA levels and the potential prognostic power in the context of therapy response. To address this concern, we have comprehensively analyzed how NEDD9 expression influences RFS and pCR in other breast cancer subtypes (ER+, TNBCs). These results were included in Supplementary Figure 2 and outlined in paragraph 3.1.
- Reviewer #2: The positive correlation between NEDD2 and Her2 needs further evaluation.
Authors: To address this concern, we continue exploring the correlation between NEDD9 and HER2 in the later stages of tumorigenesis using the mouse model described in this manuscript. The results corroborate our current findings and will be published as a part of another manuscript currently in preparation with an in-depth analysis of mechanisms and signaling outcomes. We hope that this is an acceptable resolution.
- Reviewer #2: From Fig 7 it is clear that NEDD9 does not require Her2 to activate the MAPK signaling and promote cell proliferation. Therefore, the association of NEDD9 to Her2 signaling does not seem relevant especially when the authors still attempt to block Her2 in Fig 6D. Considering that Her2 required NEDD9 to activate its downstream signaling, the focus of this manuscript should have been on targeting NEDD9.
Authors: To address this concern, we would like to emphasize that our results in Figure 6 show dual targeting of NEDD9 (shRNAs) and HER2 (lapatinib) in HER2+ established cell lines. In Figure 6D, the depletion of NEDD9 significantly sensitized HER2+ cell lines to inhibition by lapatinib. Figure 7 shows the experiments on MCF10A, non-malignant, immortalized human mammary cells. This cell line has no overexpression of EGFR, other HER2 partners, or ligands needed to activate HER2. These findings are in line with the published reports stating the need for additional changes to activate HER2 [5]. By increasing proliferation, NEDD9 creates permissive conditions for such additional changes to accrue and thus promote early HER2-driven tumorigenesis. The experimental window of 10-14 days used for acini formation and WBs is insufficient to allow such changes to happen. In this regard, the mouse study was critical to test our hypothesis and provide experimental evidence of early incidence of benign lesions (this manuscript) and later tumors (next manuscript). The NEDD9 does not require HER2 to activate the MAPK signaling but allows for setting the stage for HER2 to form preneoplastic lesions. Further clarifications have been made to the text to emphasize these findings.
- Reviewer #2: In line with the above, the authors must test pharmacological drugs targeting NEDD9 (if available) in Her2+ breast cancer models that are sensitive and resistant to anti-her2 treatment.
Authors: We agree with Reviewer, but no NEDD9-targeting drugs are currently available. It is challenging to drug NEDD9 since most of the protein is highly unstructured by design to serve as an adaptor molecule for multiple signaling hubs. We are looking into developing new anti-NEDD9 compounds based on the PROTAC technology that enables the degradation of NEDD9. This is especially critical in light of our current findings of its critical role in HER2-targeting therapy response. Also, we wanted to emphasize that we utilized HER2+ cell lines that are sensitive and resistant to anti-Her2 treatment in Figure 6D. This might at least partly address the Reviewer's concern. Being that there are no targeted NEDD9 inhibitors, we utilized shRNA as an alternative approach for NEDD9 inhibition.
- Prat, A., et al., Characterization of cell lines derived from breast cancers and normal mammary tissues for the study of the intrinsic molecular subtypes. Breast Cancer Research and Treatment, 2013. 142(2): p. 237-255.
- Muthuswamy, S.K., et al., ErbB2, but not ErbB1, reinitiates proliferation and induces luminal repopulation in epithelial acini. Nature Cell Biology, 2001. 3(9): p. 785-792.
Reviewer 3 Report
The manuscript by Purazo et al. provides a significant amount of additional information about the potential role of Nedd9 in the context of HER2+ breast cancers. There is a good grounding for significance in TMA results and characterization of a new conditional mouse model with interesting results on breast morphology and cooperation with HER2 to increase tumorigenesis and resistance to targeted therapeutic. The significant problems with the manuscript fall into two groups.
The more serious problems come from the development of the rationale and the discussion/conclusions. The context of the Nedd9 work is presented both in terms of tumor imitation/tumorignenesis and in reduction of relapse-free survival. While these two effects are not mutually exclusive, they are significantly different problems and the manuscript does not attempt to put those into context in the Discussion. [The Discussion is weak - mostly a recapitulation of the Results]. There is a conclusion drawn (including in the Title) about the potential of Nedd9 as a novel prognostic marker and claim about the potential and importance of early detection (line 50) but it is not at all clear how that could work when you would presumably need a clinical sample to be biopsied.
There are some issues with the description of the results. The data on 2D proliferation (Fig. 7B) is reported as increasing proliferation (line 643) but the data actually show that the proliferation is identical for the first 72 hours. These data look more like the Nedd9 cells gain the ability to continue proliferating after the control cells slow down (getting to confluence)? The data on 3D proliferation (Fig. 7D) look very impressive except for how tiny the error bars are (appear to be about 1 cell or less), which suggests that the SEM has been generated using a very large "n" value to reduce the apparent error size. The description of the method (lines 359-361) suggests that there were only 3 independent experiments but perhaps the calculation was done using the total number of acini counted (50 or more?).
Other problems in the manuscript are due to some problems with grammar and confusion. E.g., lines 85-86 say that "increased Nedd9 expression tightly correlates with... anti-HER2 therapy response" but the data seem to show the opposite (see line 94-95); sentences missing verbs (lines 87-88 & 561-563) or connected with a comma (lines 737-739). The use of the definite article ("The") is very inconsistent; wrong tense ("overexpress") in line 641. Inconsistent use of superscripts is confusing (e.g., lines 229 & 340). The clone designation is only given for the anti-Nedd9 antibody - there is insufficient information provided about the other primary antibodies for the work to be repeated. What does "... early lesions at the ~16 weeks" (line 551) mean?
Author Response
MS# Cancers-2150513. Revision.
Date: January 19th 2023
NEDD9 overexpression causes hyperproliferation of luminal cells and cooperates with HER2 oncogene in tumor initiation: a novel prognostic marker in breast cancer.
Author's Response to Reviewer's Critique:
We appreciate the reviewers and editors for their thorough review of our manuscript and for providing constructive feedback on its critical findings. The text has undergone scientific and English editing to improve the reading and presentation of our work. The changes are easy to track in the text to streamline the review process. Below are point-by-point responses to the raised concerns. We hope the revised manuscript addresses all the points raised and will be acceptable for publication.
Reviewer #3 (Comments to the Author):
Reviewer #3 The more serious problems come from the development of the rationale and the discussion/conclusions. The context of the Nedd9 work is presented both in terms of tumor imitation/tumorignenesis and in reduction of relapse-free survival. While these two effects are not mutually exclusive, they are significantly different problems and the manuscript does not attempt to put those into context in the Discussion. [The Discussion is weak - mostly a recapitulation of the Results].
Authors: We agree with Reviewer that, conceptually, tumor initiation and RFS might be significantly different problems but arise from the same source. Following Reviewer's suggestion, we discussed these points in the revised Discussion to provide better context to our findings. We hope that the revised Discussion will be acceptable in strengthening the rationale and conclusions. The following text was added to the Discussion:
"The increase in stem-like population, in combination with increased proliferation, might explain the increase in the incidence of benign lesions upon NEDD9 overexpression. NEDD9 promotes stemness across various cancers[6-8]. Cancer stem/initiating cells were found in the patient's biopsies and significantly influenced the risk of relapse [9, 10]. In this context, NEDD9-dependent increases in initiation and RFS might be connected. Similarly, the role of NEDD9 in EMT previously documented by multiple reports links it the resistance to the therapies. Though no NEDD9-targeting drugs are currently available, the inhibitors for some of its downstream effectors, like AURKA and FAK are available and shown to be effective [11, 12]. "
Thus, elucidating the molecular mechanisms associated with NEDD9-driven anti-HER2 drug resistance is crucial for developing more effective drugs. Here we highlighted the mechanisms exploited by NEDD9 via activation of AURKA that might aid in overcoming targeted therapies escape. Finally, novel NEDD9-targeting approaches shown in our study provide evidence of their effective preclinical applications.
Reviewer #3 There is a conclusion drawn (including in the Title) about the potential of Nedd9 as a novel prognostic marker and claim about the potential and importance of early detection (line 50) but it is not at all clear how that could work when you would presumably need a clinical sample to be biopsied.
Authors: To address this concern, we have added to discussion lines 785-788 to clarify the potential intervention window. We agree that biopsies from benign lesions, including DCIS, would be ideal for strengthening our conclusions and making recommendations for clinical intervention. We recently initiated a retrospective study that will include early stages to determine the influence of NEDD9 expression levels on the therapy responses in HER2+ BCs at our clinic at WVU Cancer Institute. We hope our findings will provide a strong foundation for future clinical decisions on diagnostics and treatment of HER2+ cancer in the context of NEDD9 expression. Unfortunately, in the provided revision time, we will not be able to assess the expression of NEDD9 in early-stage BCs. Furthermore, the DCIS biopsies, while it would be a valuable resource, are sparse and difficult to procure to enable comprehensive study. We have used the best available resources to investigate this topic and bring this to the attention of the basic and clinical researchers to promote more research on this subject.
Reviewer #3 There are some issues with the description of the results. The data on 2D proliferation (Fig. 7B) is reported as increasing proliferation (line 643) but the data show that the proliferation is identical for the first 72 hours. These data look more like the Nedd9 cells gain the ability to continue proliferating after the control cells slow down (getting to confluence)?
Authors: Excellent point. We agree that significant changes in proliferation were observed after 72h. To address this concern, we added discussion. "NEDD9 cells have enhanced ability to continue proliferation even if confluence is achieved. Mechanisms such as contact inhibition normally restrict proliferation in MCF10A cells. Based on our findings, NEDD9-overexpressing cells were able to overcome such limitations. Given the known role of NEDD9 in integrin signaling, adhesion dynamics, and proliferation, the increase in integrin activation in combination with adhesion disassembly leads to increased proliferation [13].
Reviewer #3 The data on 3D proliferation (Fig. 7D) look very impressive except for how tiny the error bars are (appear to be about 1 cell or less), which suggests that the SEM has been generated using a very large "n" value to reduce the apparent error size. The description of the method (lines 359-361) suggests that there were only 3 independent experiments but perhaps the calculation was done using the total number of acini counted (50 or more?).
Authors: We revised the description in material and methods and figure legend. We confirm that the total number of acini used for quantification was 50. The phenotype shown in Figure 7D was consistent and did not require excessive quantification.
Reviewer #3 Other problems in the manuscript are due to some problems with grammar and confusion. E.g., lines 85-86 say that "increased Nedd9 expression tightly correlates with... anti-HER2 therapy response" but the data seem to show the opposite (see line 94-95);
Authors: This was corrected. The following sentence was incorporated: "increased Nedd9 expression tightly correlates with decreased anti-HER2 therapy response".
Reviewer #3 sentences missing verbs (lines 87-88 & 561-563).
Authors: The verbs were introduced and verified via grammar editing tools.
Reviewer #3 or connected with a comma (lines 737-739).
Authors: The authors were not sure what exact sentence had been referred to in this section. We have verified all sentences between 737-739 for grammatical errors and sentence structure. We have corrected current sentences in this part of the manuscript as recommended by a professional editor. Please let us know if we are still missing something by highlighting the problematic phrasing or errors.
Reviewer #3: The use of the definite article ("The") is very inconsistent;
Authors: We rely on professional editing services to address these concerns. We hope that the revised manuscript was improved to address this concern.
Reviewer #3: wrong tense ("overexpress") in line 641.
Authors: We were not able to locate the wrong tense 'overexpress" verb in line 641. It is possible that during the figure montage, the lines were displaced. We found one wrong instance of the verb "overexpress" in lines 624-25, and it was corrected.
Reviewer #3: Inconsistent use of superscripts is confusing (e.g., lines 229 & 340).
Authors: We apologize for this oversight. The confusing superscripted text was not found in or around lines 229 and 340. If possible, please copy and paste the text to be corrected. We would be happy to attend to this and use acceptable special characters designated instead of superscripts.
Reviewer #3: The clone designation is only given for the anti-Nedd9 antibody - there is insufficient information provided about the other primary antibodies for the work to be repeated.
Authors: We have revised the text as recommended to address this concern. Additionally, we have added additional information on antibodies used in the study.
What does "... early lesions at the ~16 weeks" (line 551) mean?
Authors: We found this sentence matching line 541 and corrected it to clarify that early pre-cancerous/benign lesions were evaluated in mice at 16 weeks of age.
- Gabbasov, R., et al., NEDD9 promotes oncogenic signaling, a stem/mesenchymal gene signature, and aggressive ovarian cancer growth in mice. Oncogene, 2018. 37(35): p. 4854-4870.
- Yue, D., et al., NEDD9 promotes cancer stemness by recruiting myeloid-derived suppressor cells via CXCL8 in esophageal squamous cell carcinoma. Cancer Biol Med, 2021. 18(3): p. 705-720.
- Little, J.L., et al., A requirement for Nedd9 in luminal progenitor cells prior to mammary tumorigenesis in MMTV-HER2/ErbB2 mice. Oncogene, 2013. 33: p. 411.
- Masciale, V., et al., The Influence of Cancer Stem Cells on the Risk of Relapse in Adenocarcinoma and Squamous Cell Carcinoma of the Lung: A Prospective Cohort Study. Stem Cells Translational Medicine, 2022. 11(3): p. 239-247.
- Marzagalli, M., et al. Cancer Stem Cells—Key Players in Tumor Relapse. Cancers, 2021. 13, DOI: 10.3390/cancers13030376.
- Hu, Z., et al., Histone deacetylase inhibitors promote breast cancer metastasis by elevating NEDD9 expression. Signal Transduction and Targeted Therapy, 2023. 8(1): p. 11.
- Kozyreva, V.K., et al., Combination of Eribulin and Aurora A Inhibitor MLN8237 Prevents Metastatic Colonization and Induces Cytotoxic Autophagy in Breast Cancer. Molecular Cancer Therapeutics, 2016. 15(8): p. 1809-1822.
- Wang, J., et al., siRNA Suppression of NEDD9 Inhibits Proliferation and Enhances Apoptosis in Renal Cell Carcinoma. Oncol Res, 2014. 22(4): p. 219-224.
Reviewer 4 Report
In this manuscript, the authors demonstrated that the overexpression of NEDD9 protein is highly cooperative with HER2 in early initiation of tumorigenesis and corelates with resistance to anti-HER2 therapy. They also investigated the mechanism of NEDD9 overexpression impact on HER2 tumorigenesis. The authors showed that NEDD9 overexpression causes proliferation of luminal epithelial cells, which provides favorable conditions for tumorigenesis. They claimed that NEDD9 can serve as a prognostic marker for therapy response, and that the deletion of NEDD9 protein might benefit the treatment of therapy resistant HER2+ breast cancers.
The rational of experimental design is well-described and the generation of conditional NEDD9 knock-in transgenic mouse model can serve as references for future studies for NEDD9 overexpression models as opposed to NEDD9 silencing models, which are extensively described in literatures. Overall, this study fills the knowledge gap of current understanding of NEDD9’s role in early-stage cancer and will be beneficial to publish on Cancers.
Specific comments on the manuscript are listed below:
1. Figure 2E, the scale bar of Cre-neu-NEDD9 should match the other two for clearer comparison.
2. Need to add citation for line 521-522.
3. Starting line 541, section 3.5. The conclusion “cooperation between NEDD9 and HER2 promotes the early initiation of tumorigenesis” was drawn too early, since MMTV-Cre-NEDD9 and MMTV-Cre-Erbb2-NEDD9 both showed significant TEB increase. This statement should be written after section 3.6.
4. The abbreviation/acronym use in the manuscript is very messy. A lot of abbreviated terms were used before it was explained or expanded. For example, AUC was only explained in line 743, but have been used throughout the whole text. Similar with DCIS, MMTV, etc. Line 682 and 709 should use “NEDD9” instead of “Nedd9” for consistence.
5. Figure 7, some"+/-" signs are not aligned.
6. The arguments in this manuscript with be further supported by citing the recent publication: Hu, Z., Wei, F., Su, Y. et al. Histone deacetylase inhibitors promote breast cancer metastasis by elevating NEDD9 expression. Sig Transduct Target Ther 8, 11 (2023). https://doi.org/10.1038/s41392-022-01221-6.
Author Response
MS# Cancers-2150513. Revision.
Date: January 19th 2023
NEDD9 overexpression causes hyperproliferation of luminal cells and cooperates with HER2 oncogene in tumor initiation: a novel prognostic marker in breast cancer.
Author's Response to Reviewer's Critique:
We appreciate the reviewers and editors for their thorough review of our manuscript and for providing constructive feedback on its critical findings. The text has undergone scientific and English editing to improve the reading and presentation of our work. The changes are easy to track in the text to streamline the review process. Below are point-by-point responses to the raised concerns. We hope the revised manuscript addresses all the points raised and will be acceptable for publication.
Reviewer #4 (Comments to the Author):
Reviewer #4 1. Figure 2E, the scale bar of Cre-neu-NEDD9 should match the other two for clearer comparison.
Authors: The scale bar difference was corrected.
Reviewer #4 2. Need to add citation for line 521-522.
Authors: To address this concern, we have added a citation for lines 521-522 (PMID: 8816486).
Reviewer #4 3. Starting line 541, section 3.5. The conclusion "cooperation between NEDD9 and HER2 promotes the early initiation of tumorigenesis" was drawn too early, since MMTV-Cre-NEDD9 and MMTV-Cre-Erbb2-NEDD9 both showed significant TEB increase. This statement should be written after section 3.6.
Authors: We agree with the Reviewer and changed the Title for this chapter to reflect the content better. The revised text is: "Cooperation between NEDD9 and HER2 promotes mammary gland branching morphogenesis." The Title of chapter 3.5 was modified and introduced after 3.6 per the Reviewer's recommendation.
Reviewer #4 4. The abbreviation/acronym use in the manuscript is very messy. A lot of abbreviated terms were used before it was explained or expanded. For example, AUC was only explained in line 743, but have been used throughout the whole text. Similar with DCIS, MMTV, etc. Line 682 and 709 should use "NEDD9" instead of "Nedd9" for consistence.
Authors: To address these concerns, we added expanded terms before abbreviated terms. The use of lowercase letters in Nedd9 is based on the approved Hugo nomenclature to distinguish between murine (Nedd9) and human (NEDD9) species[14].
- Sundberg, J.P. and P.N. Schofield, Commentary: Mouse Genetic Nomenclature: Standardization of Strain, Gene, and Protein Symbols. Veterinary Pathology, 2010. 47(6): p. 1100-1104.
Round 2
Reviewer 2 Report
The revised manuscript brought some clarifications on the NEDD9-associated mechanism. While NEDD9 expression is not limited to the Her2-positive breast cancer subtype, the authors demonstrated potential therapeutic benefits. Still, several weaknesses remain: (i) lack of therapeutic-resistant Her2 models, (ii) lack of high-quality NEDD9 antibodies. Overall, however, this study may lay the foundation for more in-depth investigations.